# Herpes simplex encephalitis due to a mutation in an E3 ubiquitin ligase

Stéphanie Bibert [1], Mathieu Quinodoz[2,3,4,15], Sylvain Perriot[5,15], Fanny S. Krebs[6,7], Maxime Jan [8], Rita C. Malta[9], Emilie Collinet[1], Mathieu Canales[5], Amandine Mathias [5], Nicole Faignart [10], Eliane Roulet-Perez[10], Pascal Meylan[11], René Brouillet[11], Onya Opota[11], Leyder Lozano-Calderon[1], Florence Fellmann [12], Nicolas Guex [8], Vincent Zoete[6,7,13], Sandra Asner[1,9,16], Carlo Rivolta [2,3,4,16], Renaud Du Pasquier[5,14,16] & Pierre-Yves Bochud [1] ✉

Encephalitis is a rare and potentially fatal manifestation of herpes simplex type 1 infection. Following genome-wide genetic analyses, we identified a previously uncharacterized and very rare heterozygous variant in the E3 ubiquitin ligase *WWP2*, in a 14-month-old girl with herpes simplex encephalitis. The p.R841H variant (NM_007014.4:c.2522G > A) impaired TLR3 mediated signaling in inducible pluripotent stem cells-derived neural precursor cells and neurons; cells bearing this mutation were also more susceptible to HSV-1 infection compared to control cells. The p.R841H variant increased TRIF ubiquitination in vitro. Antiviral immunity was rescued following the correction of p.R841H by CRISPR-Cas9 technology. Moreover, the introduction of p.R841H in wild type cells reduced such immunity, suggesting that this mutation is linked to the observed phenotypes.

Herpes simplex virus type 1 (HSV-1) is a ubiquitous neurotropic human alpha herpesvirus usually causing mild mucocutaneous lesions[1]. In rare circumstances, HSV-1 can reach the central nervous system (CNS) via the olfactory bulb or the trigeminal nerve, causing life-threatening herpes simplex encephalitis (HSE)[1,2]. HSE occurs at a rate of 2–4 per million individuals per year[3] with a bimodal age distribution that includes children (at ages of 6 months to 3 years), mainly as a result of a first infection in life[4], and adults over the age of 50 years, as a result of viral reactivation. Antiviral treatments such as acyclovir have reduced HSE mortality from about 75% to about 20%, but important sequelae with significant and life-long neurological impairment are common[5–7].

The reasons why HSV-1 causes such a devastating disease in a very limited number of individuals have been unclear for a long time. Genetic studies have shown that childhood HSE results from

[1]Infectious Diseases Service, Department of Medicine, University Hospital and University of Lausanne, Lausanne, Switzerland. [2]Institute of Molecular and Clinical Ophthalmology Basel (IOB), Basel, Switzerland. [3]Department of Ophthalmology, University of Basel, Basel, Switzerland. [4]Department of Genetics and Genome Biology, University of Leicester, Leicester, UK. [5]Department of Clinical Neurosciences, Laboratory of Neuroimmunology, Neuroscience Research Centre, University Hospital and University of Lausanne, Lausanne, Switzerland. [6]Department of Oncology UNIL-CHUV, Computer-Aided Molecular Engineering, University of Lausanne, Lausanne, Switzerland. [7]Ludwig Institute for Cancer Research, Lausanne, Switzerland. [8]Bioinformatics Competence Center, University of Lausanne, Lausanne, Switzerland. [9]Pediatric Infectious Diseases and Vaccinology Unit, Woman-Mother-Child Department, University Hospital and University of Lausanne, Lausanne, Switzerland. [10]Department of Pediatrics, Child Neurology Unit, University Hospital and University of Lausanne, Lausanne, Switzerland. [11]Institute of Microbiology, University Hospital and University of Lausanne, Lausanne, Switzerland. [12]The ColLaboratory, University of Lausanne, Lausanne, Switzerland. [13]Molecular Modelling Group, SIB Swiss Institute of Bioinformatics, Lausanne, Switzerland. [14]Department of Clinical Neurosciences, Service of Neurology, University Hospital and University of Lausanne, Lausanne, Switzerland. [15]These authors contributed equally: Mathieu Quinodoz, Sylvain Perriot. [16]These authors jointly supervised this work: Sandra Asner, Carlo Rivolta, Renaud Du Pasquier. ✉e-mail: Pierre-Yves.Bochud@chuv.ch

monogenic inborn errors of immunity, leading to uncontrolled primary infection or reactivation[8], most of which with a weak clinical penetrance[9,10]. While a number of pattern recognition receptors are involved in the innate immune detection of HSV-1, these studies have identified germline loss-of-function mutations of genes specifically involved in the Toll-like receptor 3 (TLR3) signaling pathway[11-17] that lead to impaired interferon (IFN) production. Indeed, human induced pluripotent stem cells (iPSCs)-derived neurons and oligodendrocytes from TLR3-deficient patients were more permissive to HSV-1 infection compared to those from controls[18]. Since individuals who develop HSE do not present infections due to pathogens other than HSVs and usually have normal antibody titers and T-cell functions[19-22], the TLR3/IFN pathway was proposed to be crucial for CNS immunity against HSV-1[23]. However, immune deficiencies involving pathways unrelated to the TLR3/IFN pathway have also been associated with impaired CNS neuron immunity against HSV-1[24-29].

In the present work, we identified a heterozygous mutation in *WWP2*, the gene encoding the WW domain containing E3 ubiquitin protein ligase 2 in a child with HSE. This genetic defect impaired IFN and interferon-stimulated genes (ISGs) production in response to both HSV-1 and TLR3 agonists by targeting the TIR-domain-containing adapter-inducing IFNB (TRIF) protein, the TLR3-adaptor, for ubiquitination.

## Results

### Clinical presentation

The patient was a 14-month-old, previously healthy girl, admitted to the emergency ward for prolonged focal seizures with bilateral propagation, occurring in the setting of microbiologically documented varicella that started 4 days earlier. Cerebrospinal fluid PCR was positive for HSV-1 and negative for VZV. Serological immunoglobulin tests were positive against both HSV (IgG > IgM) and VZV (IgM > IgG). Cerebral MRIs revealed left temporal edema, which progressively evolved towards extensive necrosis of the left temporal lobe and focal epilepsy.

### Identification of a heterozygous *WWP2* missense mutation

We analyzed the genome of the patient and her parents by exome sequencing and a customized computer pipeline aimed at detecting rare DNA mutations. A total of 231,859 variants were identified in the proband, among which 155 were retained after filtering for quality, allele frequency (AF, <0.1%, gnomAD version v2.1.1)[30], and impact at the protein level (missense, stop-gain or canonical splicing, Fig. 1A, Supplementary Table 1). The 155 variants were all at the heterozygous state, including 5 heterozygous compound variants. Those were further selected based on deleteriousness (CADD[31] and MutScore[32]) and/or corresponding inheritance (DOMINO[33]) prediction, leaving 4 potentially deleterious variants in 3 genes (2 compound heterozygous variants in 1 gene and 2 otherwise heterozygous variants predicted to be associated with a dominant phenotype in 2 genes). The gene carrying heterozygous variants and one of the 2 genes carrying otherwise heterozygous variants were previously associated with genetic disorders, but both were unrelated to immunity. The last variant was found in WW Domain containing E3 Ubiquitin protein ligase 2 gene (*WWP2*), a gene that was not previously associated with a genetic disorder. Since WWP2 was shown to downregulate the TLR3/IFN pathway[34], which is essential in the immune response against HSV-1, it was considered our primary target.

This WWP2 variant consisted in a guanine-to-adenine heterozygous nucleotide replacement (NM_007014.4:c.2522G > A), which resulted in the substitution of an arginine residue for a histidine residue at amino acid position 841 in the Homologous to the E6-AP Carboxyl Terminus (HECT) catalytic domain (p.R841H, Fig. 1B, C). The p.R841H variant has been detected with an allelic frequency of $1.2 \times 10^{-5}$ among -125,000 controls (gnomAD version v2.1.1) and was predicted to be deleterious by 14 out of 18 pathogenicity scores including CADD.

Furthermore, the variant was classified as likely pathogenic by MutScore with a score of 0.933 (maximum is 1.0), and arginine at position 841 appeared to be conserved in all of the 46 vertebrate WWP2 orthologues we could analyze as well as in 38 paralogs (Supplementary Figs. 1, 2). The patient was the only known member of this family to have developed HSE. Direct Sanger sequencing of this DNA region in members of her family revealed that her mother, sister, and brother also carried the heterozygous p.R841H variant. The mother and the sister had HSV-1-specific serum antibodies, whereas her brother was seronegative (Fig. 1D).

### Molecular modeling analysis of the *WWP2* missense mutation

The WWP2 HECT domain consists of a large N-lobe that interacts with E2 proteins, linked to a smaller C-lobe, which catalyzes ubiquitin transfer (Fig. 2A). Arg841 is in a strand of the C-terminal part of the HECT domain, close to Cys838, namely the catalytic site within a network of hydrogen bonds and ionic interactions, mainly involving charged residues (Arg767, Asp824, His836, Asp843, Fig. 2B). Due to the aliphatic part of its side chain, Arg831 can also make hydrophobic interactions with Cys822, Phe839 and His836. In addition, Phe839 can interact with Arg841 via a cation-π interaction upon conformational changes accessible via thermal fluctuations. Without directly interacting with Arg841, Glu645, and Arg834 also contribute to the tightness of the region through ionic interactions. The substitution of an arginine residue for a histidine residue (p.R841H), much smaller and with environment-dependent protonation states, could compromise the structural integrity of the region by impacting the wild-type interaction network.

### Energetic impact of the *WWP2* missense mutation

The changes of the folding free energy, ΔΔG were calculated using FoldX[35] for each crystal structure of human HECT domains sharing at least 50% identity with the HECT domain of WWP2 (see methods for details). For each of them, without exception, the replacement of arginine by histidine at position 841 led to energetic perturbations and higher ΔΔG values (Supplementary Fig. 3), meaning that it destabilizes the structure of the region. This is in line with the structural analysis. Altogether, the disruption of the polar interaction network might lead to a more flexible region and could facilitate the accessibility of the ubiquitination site at position 838 (Cys838).

### Transient gene expression

To characterize the p.R841H variant in vitro, both wild-type (WT, *WWP2* p.R841) and mutant (*WWP2* p.R841H) cDNAs were cloned and expressed in TLR3-expressing human embryonic kidney (HEK) 293 T cells. The expression of WWP2 in cell lysates was analyzed by western blotting (Fig. 2C). The p.R841H form of *WWP2* was expressed at the same level as the WT form.

### Impaired PBMCs responses to Poly(I:C) and HSV-1

The mRNA production of IFNβ and ISGs (ISG56, MX1, and IFIT2) was then measured in PBMCs from individuals carrying the WWP2 p.R841H variant (patient with HSE and 3 family members) and from controls (patient's father and 7 unrelated individuals) after stimulation with Poly(I:C). PBMCs from individuals carrying the WWP2 p.R841H variant expressed significantly lower mRNA levels of IFNβ and IFIT2 2 h after poly(I:C) stimulation (Supplementary Fig. 4A, B). ISG56 tended to be less expressed in PBMCs from p.R841H carriers compared to those from WT individuals (Supplementary Fig. 4C). In contrast, PBMCs from individuals carrying the WWP2 p.R841H expressed similar levels of MX1 and TNFα than those from WT controls (Supplementary Fig. 4D, E). Of note, the baseline mRNA expression of TLR3, TRIF, and WWP2 was similar among p.R841H carriers compared to WT (Supplementary Fig. 4G), showing that this impaired response was not due to decreased expression of genes implicated in TLR3 signaling.

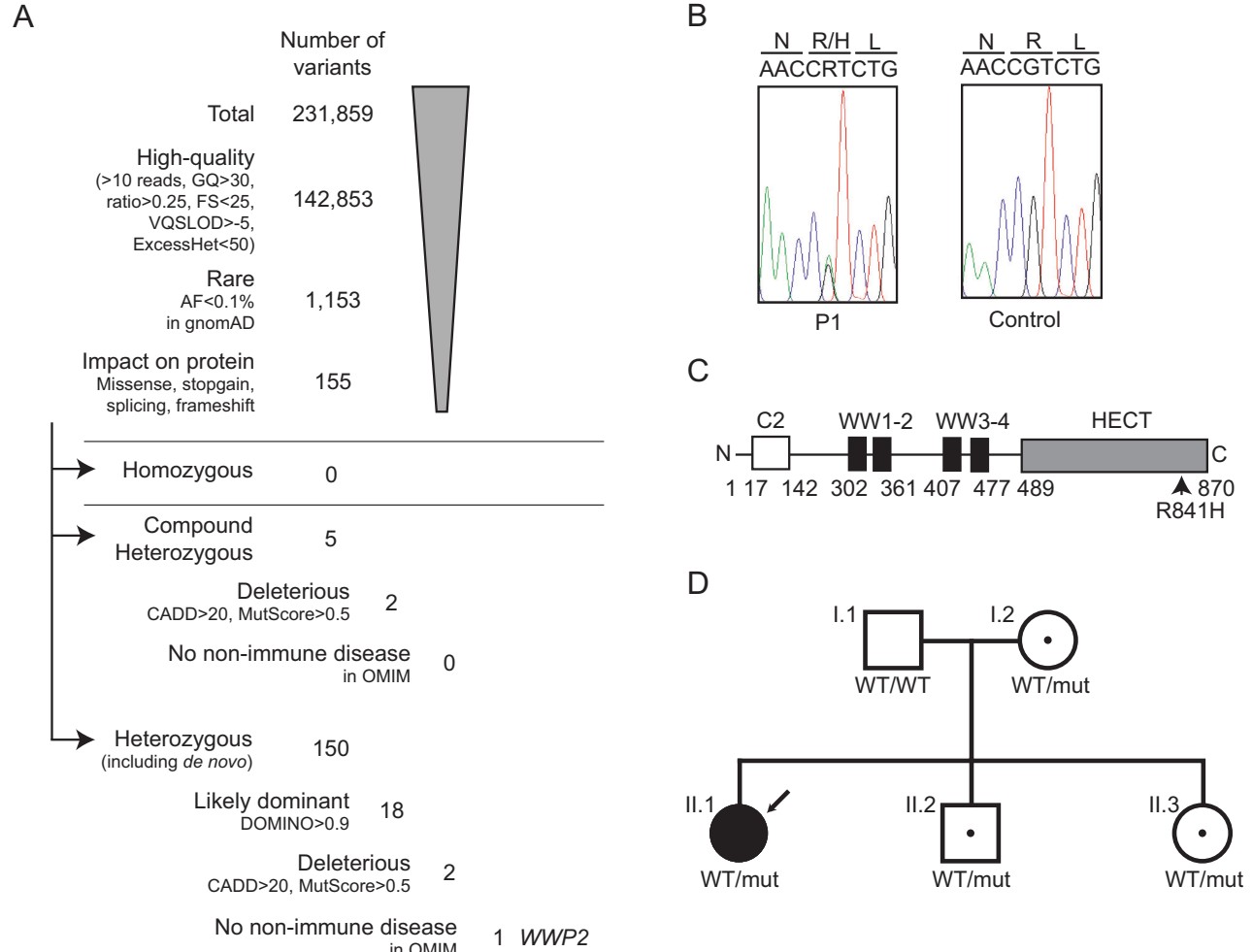

**Fig. 1 | p.R841H WWP2 variant. A** Filtering process of the variants identified by whole exome sequencing. **B** Sanger sequencing profiles for the WWP2 p.R841H mutation in genomic DNA from the patient (P1) and her father (as a control). **C** Schematic representation of WWP2, featuring the different domains as well as the location of the missense variant p.R841H. WWP2 is composed of well-defined domains comprising the C2, WW, and Homologous to the E6-AP Carboxyl Terminus (HECT) domains. **D** Family pedigree with allele segregation. The patient is indicated by the arrow sign, whereas "mut" refers to the p.R841H allele and "WT" to a wild-type allele. GQ genotype quality, FS Fisher strand, VQSLOD variant quality score log odds ratio, ExcessHet Phred-scaled $p$-value for exact test of excess heterozygosity, AF allele frequency, gnomAD genome aggregation database, CADD combined annotation dependent depletion.

In order to further characterize the impact of the p.R841H variant, we measured viral levels (copies/ml) in the supernatant of PBMCs from WWP2 p.R841H carriers and controls 18 h after HSV-1 infection (Supplementary Fig. 4F). Viral levels were significantly increased in p.R841H carriers compared to those from controls ($P < 0.0001$). Although poly(I:C) stimulation in PBMCs is not specific to TLR3, these data suggest that p.R841H carriers have a lower capacity to establish IFN responses after poly(I:C) stimulation at early time points which leads to a reduced control of HSV-1 infection.

**Impaired iPSCs-derived neuronal cells responses to TLR3 agonist and HSV-1**

To test whether the WWP2 p.R841H was associated with impaired TLR3 signaling in the CNS, we used brain-specific cell differentiation technologies from human induced pluripotent stem cells (iPSCs). Briefly, iPSCs were derived from the patient, her family members, and three unrelated donors by reprogramming erythroblasts (Fig. 3A). Generated iPSCs exhibited normal karyotype, expressed pluripotency markers and were capable to differentiate into cell types from all three germ layers (Fig. 3B, C). Three additional previously described iPSCs from WWP2 WT individuals were also used as controls[36]. Human iPSCs were then differentiated into non-hematopoietic CNS-resident cells,

comprising neural stem cells/neural precursor cells (NPCs), neurons, and astrocytes[37]. The level of maturation of each cell type, specifically neurons and astrocytes, was assessed by the expression of selected marker genes and compared to the ones of NPCs (Fig. 3D, E).

We then focused on the time-dependent expression of IFNβ, ISG56, and MX1 mRNA in different cell types. The expression of IFNβ, ISG56, and MX1 mRNA after Poly(I:C) stimulation was reduced in iPSCs-derived NPCs (Fig. 4A) and neurons (Fig. 4B) from WWP2 p.R841H carriers compared to those from WT controls. In contrast, iPSCs-derived astrocytes produced normal amount of IFNβ, ISG56, and MX1 mRNA after Poly(I:C) stimulation as shown by comparison with controls (Fig. 4C, F). As a control, the expression of IFNβ was not reduced in p.R841H carriers compared to WT controls when iPSCs-derived NPCs (Fig. 4D) and neurons (Fig. 4E) were stimulated with LPS (TLR4 agonist), CpG (TRL9 agonist) and R848 (TLR7/8 agonists), suggesting that reduced IFNβ in p.R841H carriers was specific to the TLR3/IFN axis. Altogether, these data indicated that reduced expression of IFNβ, ISG56, and MX1 mRNA after stimulation with Poly(I:C) relied on the WWP2 p.R841H mutation, both on NPCs and neurons, but not in astrocytes.

In order to provide a comprehensive analysis of the impact of WWP2 p.R841H on the TLR3 pathway, we performed RNA sequencing in neurons from both WWP2 p.R841H carriers and controls under basal

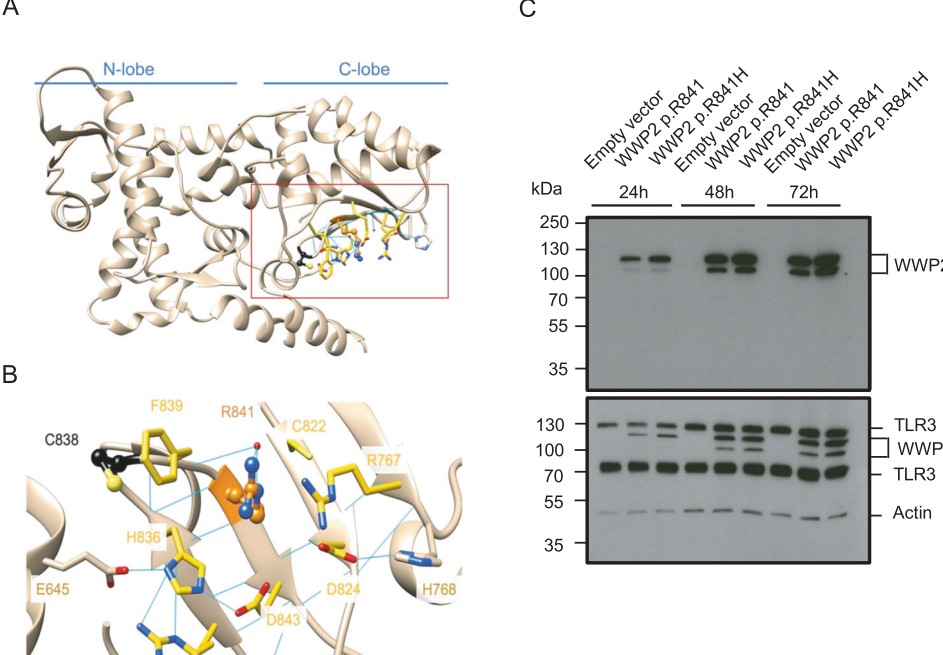

**Fig. 2 | Molecular modeling and expression analysis of p.R841H WWP2 variant.**
**A** Overview of the HECT domain structure and its two main lobes. **B** Zoom on Arg841 and its neighbors. Arg841 and Cys838 are respectively shown in orange and black ball and stick. The residues involved in the ionic and hydrogen bond network are shown in sticks. Residues with carbons colored in yellow are within 5 A from Arg841 (pdb id:4y07). **C** The expression of p.R841 and p.R841H WWP2 protein was determined by SDS-PAGE/immunoblotting under reducing conditions in cell lysates of TLR3-expressing 293 T cells after different times of transfection. Results represent 1 experiment among 3. TLR3 FL TLR3 full length.

conditions and after 3 h poly(I:C) stimulation. Alignment metrics are detailed in the Supplementary Table 3. Genes downstream NF-kB in the canonical pathway (Kyoto Encyclopedia of Genes and Genomes, KEGG has06604) as well as genes downstream AP-1 (hsa04668) had a lower expression in poly(I:C) stimulated neurons from WWP2 p.R841H carriers compared to WWP2 p.R841 neurons (false discovery rate [FDR] adjusted $P = 3.02E\text{-}02$ and $P = 3.55E\text{-}05$, respectively, Supplementary Figs. 5 and 6, Supplementary Table 4). In contrast, no significant difference was found in genes of the NF-kB non-canonical and atypical pathways (both $P = 4.02E\text{-}01$).

In order to further characterize the impact of the WWP2 p.R841H variant, we measured viral levels by qPCR (copies/ml) in both NPCs (Fig. 5A), neurons (Fig. 5B), and astrocytes (Fig. 5C) from WWP2 p.R841H carriers and controls after HSV-1 infection. Viral levels were significantly increased in NPCs and neurons from p.R841H carriers compared to those from controls. A treatment of NPCs and neurons with recombinant IFN-α2b prior to HSV-1 infection did not impact on viral level in NPCs and neurons from controls but restricted viral level in NPCs and neurons from WWP2 p.R841H carriers to a level similar to that obtained in NPCs from controls. In astrocytes from p.R841H carriers, viral levels were similar to those obtained in astrocytes from controls with or without an IFN-α2b pre-treatment. Since viral levels by qPCR do not necessarily reflect the amount of reproductive viral particles, those were measured (Pfu/ml) in neurons at different time-points (12 h, 24 h, and 48 h) after HSV-1 infection (Fig. 6A). Neurons from p.R841H carriers exhibited higher viral replication rates at all time-points, compared to those controls. A treatment of neurons with recombinant IFN-α2b prior to HSV-1 infection restricted viral replication in neurons from WWP2 p.R841H carriers to a level similar to that obtained in neurons from controls (Fig. 6B).

**TRIF ubiquitination is increased by WWP2 p.R841H compared to WWP2 p.R841**
Since WWP2 p.R841H was predicted to facilitate the accessibility of the ubiquitination site at position 838 (Cys838), we performed an in vitro

assay to compare the level of TRIF ubiquitination by WWP2 variants. These experiments showed that WWP2 p.R841H induces increased TRIF ubiquitination compared to WWP2 p.R841 (Fig. 7).

**CRISPR-mediated genome editing restores WWP2 function in the mutant and aborts it in the WT**
We used the CRISPR-Cas9 technology to generate wild type iPSCs from the patient, naturally carrying p.R841H, and to establish mutant iPSCs from the patient's father, who carried no mutations at this site. We then differentiated them into neurons to estimate the TLR3 response. Impaired Poly(I:C) responsiveness in the patient's neurons (p.R841H) was rescued by expression of only wild type form of WWP2 through CRISPR-Cas9 genome editing (p.H841R), with a level of induction of IFNβ mRNA similar to that of WT controls (p.R841) (Fig. 8A). Upon poly(I:C) stimulation, neurons from the patient's father harboring the heterozygous WWP2 p.R841H presented a lower level of induction of IFNβ mRNA compared to that obtained in previously WT neurons from the patient's father (p.R841), but similar to that of WWP2 p.H841 carriers (Fig. 8B). Viral levels were decreased in neurons from the patient expressing the wild type WWP2 p.R841 compared to those from the patient harboring the heterozygous WWP2 p.R841H (Fig. 8C) as well as the viral replication (Fig. 8G). Similarly, viral levels (Fig. 8D) and viral replication (Fig. 8H) were both increased in neurons from the patient's father harboring the heterozygous WWP2 p.R841H compared to those from the patient's father expressing the wild type WWP2 p.R841. A treatment of neurons with recombinant IFN-α2b prior to HSV-1 infection did not impact on viral replication in neurons from individuals expressing the wild type WWP2 p.R841, but restricted viral replication in neurons from WWP2 p.R841H carriers to a level similar to that obtained in neurons from WWP2 p.R841 individuals (Fig. 8C–F). Altogether, these data showed that WWP2 p.R841H carriers had a lower capability to control HSV-1 infection in neuronal cells compared to controls due to an impaired IFN production.

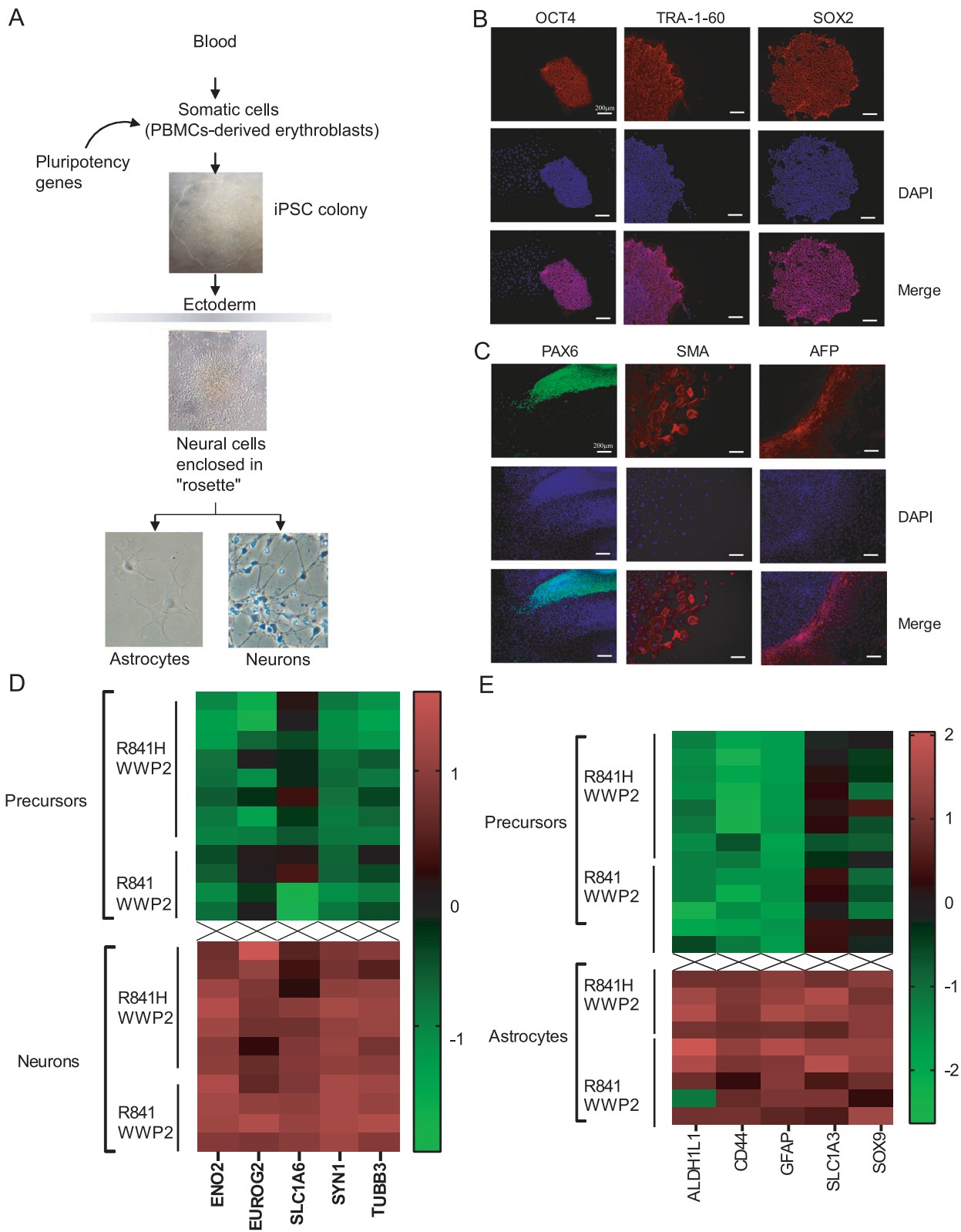

**Fig. 3 | Reprogrammation, derivation, and CNS cells differentiation.**
**A** Schematic diagram of the reprogrammation and differentiation protocols used.
**B** Immunocytochemistry analysis of iPSCs colonies revealed the expession of pluripotent markers OCT4, TRA-1-60, and SOX2. **C** Immunocytochemistry analysis of the ability of iPSCs-derived embryonic bodies to differentiate into the 3 germ layers. PAX6, SMA, and AFP are markers for ectoderm, mesoderm, and endoderm respectively. Results represent the analysis of 1 iPSCs colony and is representative of results obtained for all iPSCs colonies (*n* = 2 per donor). **D** Characterization of neurons. Compared to neuronal precursors, neurons exhibit increased expression of typical neuronal markers assessed by qPCR. **E** Characterization of astrocytes. Compared to neuronal precursors, astrocytes exhibit increased expression of typical astrocyte markers, assessed by qPCR.

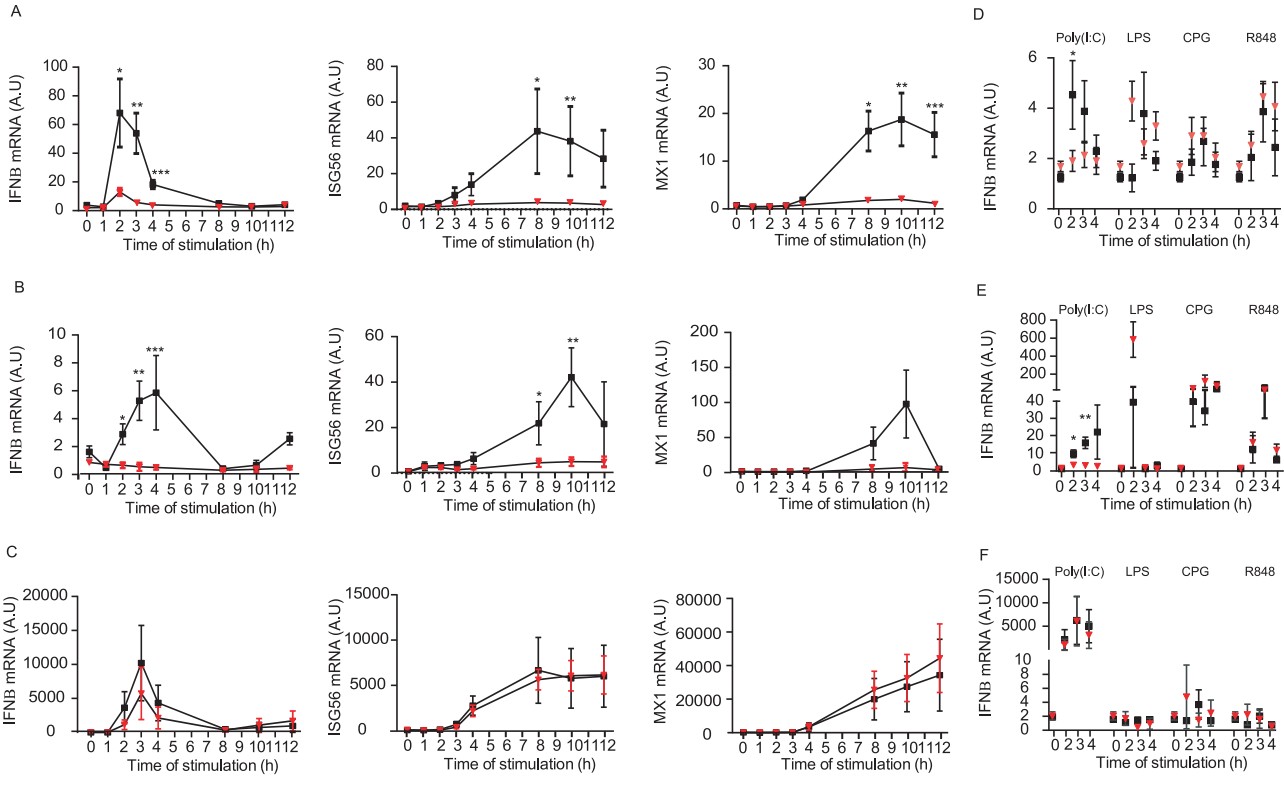

**Fig. 4 | WWP2-dependent IFN responses in iPSCs-derived neural stem cells (NSCs).** Induction of IFNβ, ISG56 and Mx1 mRNA after different time of Poly(I:C) stimulation in neural precursor cells (**A**), neurons (**B**) and astrocytes (**C**) from WWP2 p.R841 individuals [black, N = 5 (**A**), N = 4 (**B**), N = 5 (**C**)] and from WWP2 p.R841H individuals [red, N = 8 (**A**, **B**), N = 4 (**C**)]. As a control, the expression of IFNβ was measured in neural precursor cells (**D**), neurons (**E**) and astrocytes (**F**) from WWP2 p.R841 individuals (black) and from WWP2 p.R841H individuals (red), stimulated with LPS [WWP2 p.R841 individuals N = 5 (**D**, **E**), N = 4 (**F**); WWP2 p.R841H individuals N = 8 (**D**, **E**), N = 2 (**F**)], CPG [WWP2 p.R841 individuals N = 5 (**D**, **E**), N = 4 (**F**); WWP2 p.R841H individuals N = 8 (**D**, **E**), N = 2 (**F**)] and R848 [WWP2 p.R841 individuals N = 5 (**D**, **E**), N = 4 (**F**); WWP2 p.R841H individuals N = 8 (**D**, **E**), N = 2 (**F**)] for 2, 3 and 4 h. A.U. stands for arbitrary unit. Results represent the mean ± standard error of 1 representative experiment among 2 (**A**, **B**), 3 (**C**), and 1 (**D–F**). Statistical analyses were performed using an unpaired two-tailed Student t-test. **A** Left *, **, *** mean for P = 0.0129, P = 0.0011 and P = 0.0001, (**A**) middle *, ** mean for P = 0.0501 and P = 0.0413, (**A**) right *, **, *** mean for P = 0.0010, P = 0.0025, P = 0.0021. **B** Left *, **, *** mean for P = 0.0026, P = 0.0006 and P = 0.0187, (**B**) middle *, ** mean for P = 0.0299 and P = 0.0022. **D** * means for P = 0.0475. **E** *, ** mean for P = 0.0121 and P = 0.0005.

## Discussion

Here, we report a heterozygous missense *WWP2* variant associated with HSE in a 14-month-old child. WWP2, also called Atrophin-1 Interacting Protein 2 (AIP2), is an E3 ubiquitin ligase, which plays a central role in protein ubiquitination, a post-translational mechanism of protein modification catalyzed by the sequential action of ubiquitin-activating (E1), ubiquitin-conjugating (E2) and ubiquitin-ligating (E3) enzymes. The human genome encodes more than 600 E3 ligases and this relatively high diversity enables their specificity in targeting protein substrates. Recently, in vitro ubiquitination assays, performed in both 293-TLR3 and A549 cells, showed that WWP2 mediates the K48-linked ubiquitination and degradation of the TLR3 adaptor TRIF following poly(I:C) stimulation. Moreover, in reporter assays, overexpression of WWP2 inhibited poly(I:C) induced transcription of *IFNB1, CCL5, and ISG15* while its knockdown increased the expression of these genes[34].

Previous studies have convincingly shown that susceptibility to HSE can be due to mutations in members of the TLR3 signaling pathway. Similar to our results, such mutations displayed a very low degree of penetrance, i.e., a substantial percentage of carriers of them did not develop disease, despite their unfavorable genotype[11–17]. This phenomenon highlights the importance of specific interactions between the environment (exposure to pathogens) and the host (immune status, co-infections) during a given phase of life, for these mutations to produce clinical consequences, while in fact they result in clear deficiencies at the molecular level, as shown for instance by cells derived from the mother and the siblings of our patient. In the present case, co-infection by both VZV and HSV-1, together with WWP2 p.R841H carriage, may have been required for the development of HSE. Intriguingly, WWP2 is not directly involved in the signaling cascade of the TLR3/IFN axis but is known to regulate this pathway through K48-linked ubiquitination of TRIF following Poly(I:C) stimulation[34]. WWP2 p.R841H carriers had a reduced expression of genes downstream NF-kB, AP-1, and IRF3, including IFNβ and ISGs, after Poly(I:C) stimulation. This was observed in both iPSC-derived NPCs and neurons, but not astrocytes, and was associated with an enhanced susceptibility to HSV-1 infection. These observations overlap with those from two children with TRIF deficiency (autosomal dominant p.S186L and autosomal recessive p.R141X), in which dermal fibroblasts stimulated with Poly(I:C) exhibited an impaired production of IFNβ as well as higher levels of HSV-1 replication[14]. The observed phenotype was linked to the WWP2 p.R841H, as allelic reversion (H841 > R) rescued the control of HSV-1 replication in iPSC-derived neurons from the patient, while the introduction of a single mutant allele (R841 > H) abolished IFN-mediated immunity against HSV-1 in iPSC-derived neurons from her wild type father. Interestingly, IFN production following LPS stimulation was conserved in both the WWP2 p.R841H and the TRIF p.S186L patients, but not in the TRIF p.R141X patient, suggesting that heterozygous mutations are sufficient to impact TRIF signaling in the

A

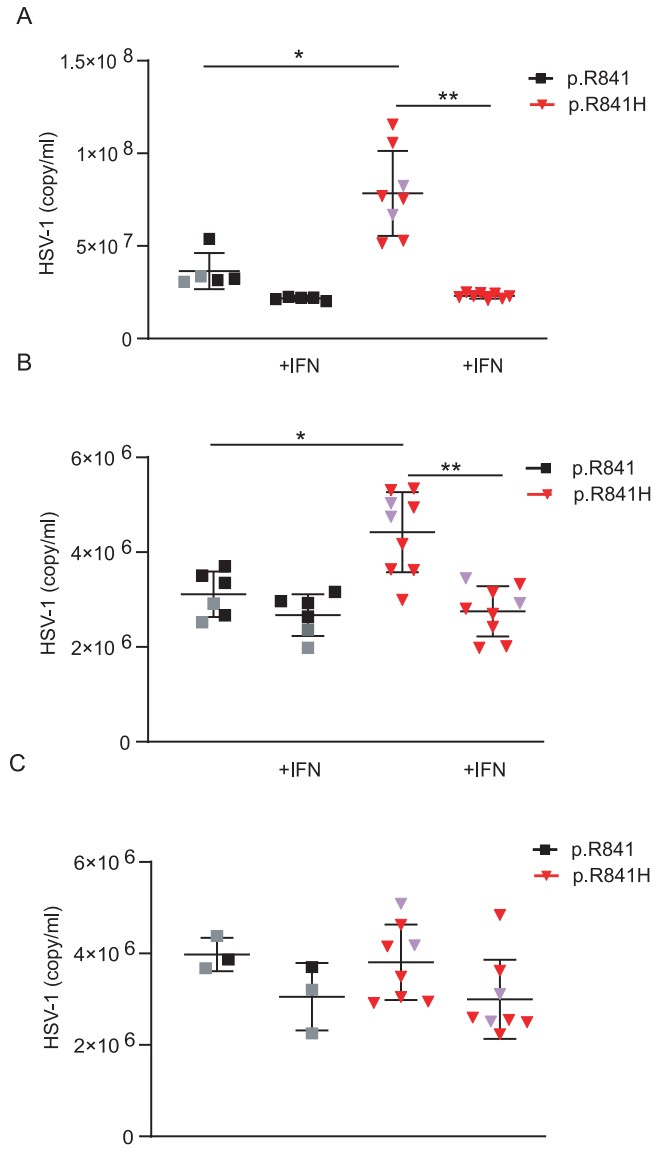

**Fig. 5 | WWP2-dependent HSV-1 genomes number in iPSCs-derived neural stem cells (NSCs).** Quantification by polymerase chain reaction of the number of viral genome in neural precursor cells (**A**), neurons (**B**) and astrocytes (**C**) from WWP2 p.R841 individuals [black, $N = 5$ (**A**), $N = 6$ (**B**), $N = 3$ (**C**)] and from WWP2 p.R841H individuals [red, $N = 8$ (**A**), $N = 9$ (**B**), $N = 8$ (**C**)] without or with an IFNα2b pre-treatment (+IFN). Results represent the mean ± standard error of 1 experiment among 2 (**A**) and 1 (**B**, **C**). Statistical analyses were performed using an unpaired two-tailed Student $t$-test. **A** *, ** mean for $P = 0.0028$ and $P < 0.0001$. **B** *, ** mean for $P = 0.0046$ and $P = 0.0001$. The purple triangle and the grey square correspond to patient with HSE and to patient's father respectively.

TLR3, but not in the TLR4, pathway, and that homozygous mutations are required to abolish TRIF signaling in both TLR3 and TLR4 pathways[14].

Interestingly, WWP2 can regulate its own catalytic activity through an intramolecular interaction mechanism involving trapping of the HECT domain[38]. Molecular modeling analyses and calculation of the change in the folding free energy indicate that the WWP2 p.R841H mutation leads to a more flexible region and may facilitate the accessibility of the ubiquitination site, thereby promoting a constitutive activation of the enzyme. This is in agreement with our in vitro assay showing an increased TRIF ubiquitination by the WWP2 p.R841H variant compared to the control.

Ubiquitination is a complex post-translational mechanism which is critical for protein homeostasis and cell signaling and regulates various cellular processes including immune responses[39]. It can modulate immune pathways by targeting for degradation key proteins involved in pathogen's recognition, such as PRRs and their adaptor molecules[34,40], their downstream signaling proteins[41,42], or transcription factors[43,44]. Ubiquitination can also act on immune functions without altering protein stability by impairing the assembly of protein complexes or by promoting the activation of specific proteins, separating them from their binding partner(s)[39,40,42,45,46].

Due to its crucial role in the TLR3-mediated signaling, the expression of TRIF must be tightly controlled as well as the expression of its HECT E3 ubiquitin ligases in order to ensure normal cell behavior and to prevent illnesses[47–49]. At a post-transcriptional level, TRIF is heavily modified and regulated by polyubiquitination, leading to the inhibition of the TLR3- and TLR4-mediated innate immune and inflammatory responses, through different mechanisms, depending on the type of E3 ubiquitin ligase. While polyubiquitination of TRIF by WWP2 results in its proteasomal degradation (36), polyubiquitination by TRIM38[34,50], TRIM8[46], or USP19 (44) are thought to impair its complex assembly without degradation.

Our study has limitations. First, the molecular modeling analysis of the mutation focuses on the free energy change and the interaction network within the active site of the HECT domain. It does not explain the activating effect of p.R841H mutation. Since WWP2 HECT domain adopts auto-inhibitory conformation, it would be interesting to test whether the p.R841H mutation activates WWP2 by releasing auto-inhibition. Molecular dynamics simulations or normal mode analysis would certainly decipher the exact effect of the p.R841H mutation. Second, although we show that the WWP2 p.R841H variant is associated with an increased TRIF ubiquitination in vitro, we do not investigate whether it enhances TRIF ubiquitination in iPSC-derived neurons, whether this leads to TRIF degradation and how it leads to change in the control of HSV-1 infection. Lastly, although genome editing experiments show significant differences on viral replication in a time-dependent manner, the limited number of genetically modified iPSCs colonies and/or time points did not allow for statistical analyses regarding IFN induction. However, introduction and rescue of the mutant allele provided consistent trends towards decreased and increased IFN production, respectively, in repeated experiments.

Altogether, our study further confirms the role of the TLR3/IFN pathway in controlling HSV-1 infection in the CNS during childhood. We show that a mutation in *WWP2* leading to increased TRIF ubiquitination in vitro is associated with a phenotype similar to what is observed in TRIF deficient patients.

## Methods
### DNA extraction
DNA from either PBMCs or whole blood was isolated using the QIAamp DNA mini kit (Qiagen) visualized by gel electrophoresis to verify quality and then quantified by Picogreen or on a NanoDrop 1000 spectrophotometer (Thermo Fischer Scientific).

### Exome sequencing
Exome was performed for the proband and her parents using the Twist Exome 2.0 capture kit (Twist Bioscience, San Francisco, USA) and Illumina NovaSeq 6000 (Illumina, San Diego, USA), resulting in sequences of 100 bases. Raw reads were mapped to the human genome reference sequence (build hg19) with BWA mem (v0.7.17). Duplicate reads were removed using Picard (v. 2.14.0-SNAPSHOT). Base quality score recalibration was performed with GATK dedicated functions and variant calling done with HaplotypeCaller (GATK, v.4.1.4.1). The variants called were annotated with a custom pipeline, adding information about effect of the variants at the protein level, frequency in various databases, conservation across species, effect on

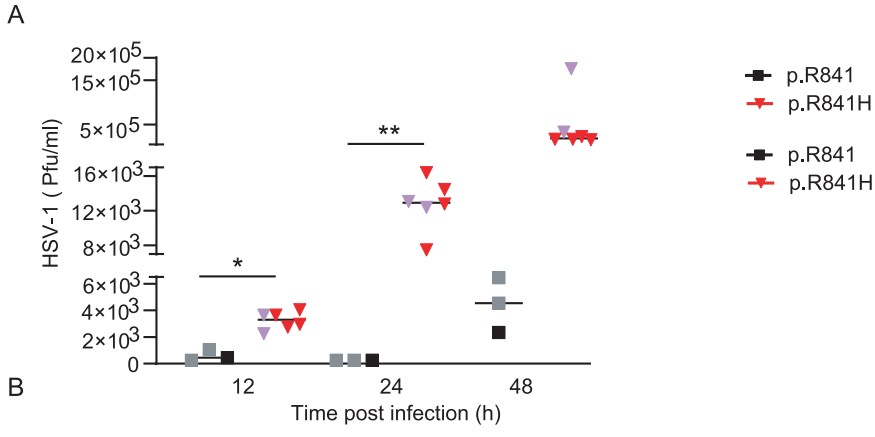

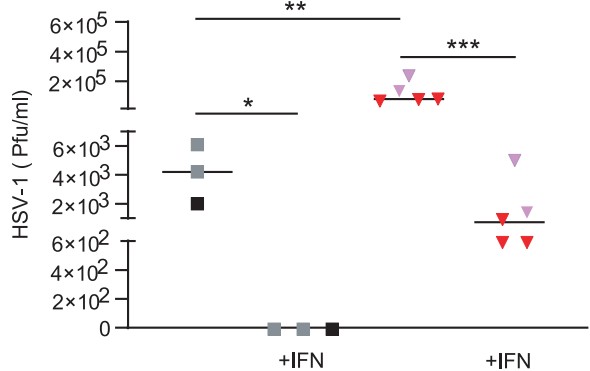

**Fig. 6 | WWP2-dependent HSV-1 replication in iPSCs-derived neurons.**
**A** Quantification of the number of plaque-forming units (pfu) by plaque-titration at different time after infection of neurons from WWP2 p.R841 individuals (black, N = 3) and from WWP2 p.R841H individuals (red, N = 6). (B) Quantification of the number of plaque-forming units (pfu) by plaque-titration at 48 h after infection of neurons from WWP2 WT individuals (black, N = 3) and from WWP2 p.R841H

individuals (red, N = 5) without or with an IFNα2b pretreatment (+IFN). Figure shows median of one experiment. Statistical analyses were performed using an unpaired two-tailed Student t-test. The purple triangle and the grey square correspond to patient with HSE and to patient's father respectively. **A** *, ** mean for P = 0.0005 and P = 0.0002. **B** *, **, *** mean for P = 0.0239, P = 0.0256 and P = 0.0039.

splicing and deleteriousness predictors, using mainly ANNOVAR. More details on the annotation and filtering can be found in Peter et al.[51].

## Molecular modeling analysis

All experimental 3-dimensional (3D) structures were extracted from the Protein Data Bank[52]. Structures and sequence alignments were analyzed using USCF Chimera version 1.13.1 visualization software[53]. FoldX version 4 was used to generate structural models of the mutant and to estimate the energetic impact of the mutation (ΔΔG values in kcal/mol)[35]. SCWRL4 was used to create models based on experimental structures, with optimized amino acid sidechain positions[54]. Protein sequences were extracted from the UniProt database[55] and multiple sequence alignments (MSAs) were performed using MUSCLE version 3.8.31[56]. All structural images were generated in-house, using USCF Chimera[53].

## Sequence conservation analysis

To study the sequence conservation among human paralogs, UniProt database[55] was used. A total of 133 human protein sequences with a HECT domain was retrieved. A multiple sequence alignment (MSA) was performed using MUSCLE[56]. Sequences covering the human WWP2 HECT region, based on UniProt delineation of human WWP2 HECT domain, were extracted and sequence identities with the human WWP2 HECT domain were calculated using UCSF Chimera[53].

38 sequences with larger than 50% sequence identity to WWP2 were selected for further analysis (Supplementary Fig. 2).

## Estimation of the energetic impact of the variant compared to the wild type structure

Only 5 experimental 3D structures of the HECT domain of WWP2 are published and publicly available from the Protein Data Bank (4y07[57], 5tj7[38], 5tj8[38], 5tjq[38], 6j1z[58]). To increase this number, we retrieved human structures of the HECT domain of proteins that share a sequence identity higher than 50% with human WWP2. 23 experimental structures were selected (Supplementary Table 2), for ITCH, NEDD4, NEDD4L, SMURF2, WWP1 and WWP2. The chains containing the HECT domain have been retained for the following steps. To further increase our sampling, we created a collection of 3D structural models based on the experimental structures, using SCWRL4[54]. SCWRL4 checks and, if needed, corrects the side chain positions of structures. Indeed, a side chain position quality depends on the electron density detected and can be doubtful in some experimental structures. FoldX was used to generate structural models of the mutant and to estimate the energetic impact of the mutation on the Arg841 environment[35]. FoldX is an efficient software for predicting mutation-dependent folding-free energy changes, whose predictive efficiency has been trained on a large mutation set[59]. The predicted energy perturbation (ΔΔG, kcal/mol) induced by the mutant is calculated. Several experimental structures were used for FoldX

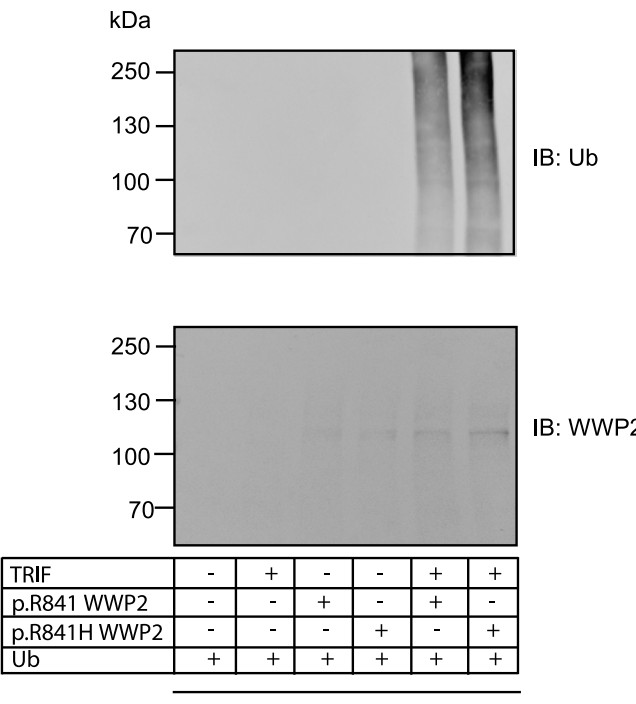

**Fig. 7 | Increased TRIF ubiquitination by WWP2 p.R841H compared to WWP2 p.R84.** TRIF, p.R841 WWP2, and p.R841H WWP2 were in vitro translated. An in vitro ubiquitination assay of TRIF was conducted by using an E1 ubiquitin-activating enzyme and UbcH5c together with either p.R841 WWP2 or p.R841H WWP2. Ubiquitin-conjugated TRIF was detected by immunoblot with an anti-ubiquitin antibody. The expression levels of WWP2 were also verified by immunoblots with an anti-WWP2 antibody. Results represent 1 experiment among 2.

calculations, to strengthen the interpretation of the predicted energy perturbation results, as the approach is sensitive to small conformational changes observed between different experimental 3D structures or accessible via thermal fluctuations.

## PBMCs isolation

Peripheral blood mononuclear cells (PBMCs) were prepared from fresh EDTA (1.6 mg/ml) blood from patient, patient's relatives, and healthy donors with written consent and approval of the Ethics committee. Briefly, whole blood diluted in PBS was overlaid above Ficoll-Paque Plus (GE Healthcare, Uppsala, Sweden), and mononuclear cells extracted by gradient density centrifugation. Viability, determined by trypan blue exclusion, was >90%.

## HSV-1 amplification and HSV-1 plaque assay

Human herpesvirus 1 strain MacIntyre used in viral neutralization studies was purchased from ATCC (ATCC® VR-539) and was grown on a 80% confluency African green monkey kidney (Vero, ATCC® CCL-81) cell line in DMEM/GlutaMax supplemented with 10% FBS. Briefly, Vero cells were infected with HSV-1 at a multiplicity of infection of 1, during at least 2 h at 37 °C. Medium was then replaced and supernatant was collected after 3–4 days incubation at 37 °C, until the appearance of cytopathic effect, and clarified at 4500× g for 30 min. HSV-1 titers were determined by a standard plaque assay on Vero cells. Briefly, serial dilutions of HSV-1 suspension were incubated on confluent mono-layers of cells during at least 2 h at 37 °C. Supernatant was then aspirated and cells washed with PBS. Cells were then incubated in DMEM containing 1% MethylCellulose and 0.33% Sodium Bicarbonate at 37 °C for several days, until the plaque formation. Medium was then aspirated and the cells were fixed and stained with a mixture of Par-aformaldehyde 2% and crystal violet 0.5% for 15 min at room temperature. HSV-1 titer was evaluated from plaque counting and expressed as plaque-forming units (PFU)/ml.

The virus preparation was aliquoted and stored at −80 °C.

## TLRs agonists, HSV-1 infection, and quantification

Primary or neural stem cells and cell lines were stimulated with 50 μg/ml poly inosine polycytidylic acid (poly(I:C)), 0.1 μg/ml LPS, 5 μg/ml R848, 8 μg/ml CpG, and 1 PFU/cell HSV-1, for various periods of time. Primary or neural stem cells were treated or not with 10'000 units/ml IFNα2b (pbl assay science) during 24 h before being infected with HSV-1 at a multiplicity of infection of 1. Between 12 and 48 h post infection, cell supernatant was collected. Total viral DNA were extracted with the MagNA Pure 96®, according to the manufacturer's instructions, and were quantified by RT-PCR[60]. The quantification of infectious virus were done by using plaque assay as detailed above.

## RNA extraction

Total RNAs were extracted, using an RNeasy Mini kit and the automated QIAcube (Qiagen, Hombrechtikon, Switzerland) from patient/patient family/healthy volunteer PBMCs and/or iPSCs-derived cells, treated or not with different stimuli including pattern recognition receptor agonists or HSV-1.

## Messenger RNA expression

Total RNAs were reverse transcribed in the presence of random primers using the QuantiTect Reverse Transcription Kit (Qiagen). The relative levels of TLR3, WWP2, TRIF, IFNB, ISG56, Mx1, IFIT2, and TNF transcripts were determined by RT-PCR, with a QuantStudio 12 K Flex Real-Time PCR system, using the Power SYBR green PCR master mix (Thermo Fisher) with primers described in Supplementary Table 5. Primers were designed using the Primer 3 software and validated by BLAST analysis. The relative levels of mRNA expression to HPRT were determined by the $2^{(-\Delta\Delta Ct)}$ method as described by the manufacturer and expressed in arbitrary units (A.U.). HPRT expression was not influenced by cell stimulation.

## RNA quality control and library preparation

RNA quality was assessed on a Fragment Analyzer (Agilent Technologies, Santa Clara, CA, USA), and all RNAs had an RNA quality number (RQN) comprised between 7.8 and 10. Library preparation and RNA-seq were performed at the Lausanne Genomic Technologies Facility, University of Lausanne, Switzerland (https://www.unil.ch/gtf). Briefly, the Truseq Stranded mRNA reagents (Illumina, San Diego, CA, USA) were used for the library preparation with 108 ng of total RNA as input. RNA-seq libraries were quantified by a fluorimetric method (QubIT, Life Technologies, Carlsbad, CA, USA). Their quality was assessed on a Fragment Analyzer. Sequencing was performed on an Illumina Novaseq 6000 for 100 cycles (single end). Sequencing data was demultiplexed using the bcl2fastq2 Conversion Software (version 2.20, Illumina). Sequencing yield was 877 M pass-filter reads with 41–61 M for each library.

## RNA sequencing data processing

Purity-filtered reads were adapters and quality trimmed with Cutadapt (v. 2.5[61]). Reads matching to ribosomal RNA sequences were removed with fastq_screen (v. 0.11.1). Reads were aligned against Homo_sapiens GRCh38 genome using STAR (v. 2.5.3a[62]). The number of read counts per gene locus was summarized with htseq-count (v. 0.9.1[63]) using gene annotation from the Homo_sapiens_GRCh38.102.gtf file. Quality of the RNA-seq data alignment was assessed using RSeQC (v. 2.3.7[64]). Statistical analysis was performed in R (R version 4.2.2). Genes with low counts were filtered out according to the rule of 1 count per million (cpm) in at least 1 sample. Library sizes were scaled using TMM normalization. Subsequently, the normalized counts were transformed to cpm values, and a log2 transformation was applied by means of the function cpm with the parameter setting prior counts = 1 (EdgeR v

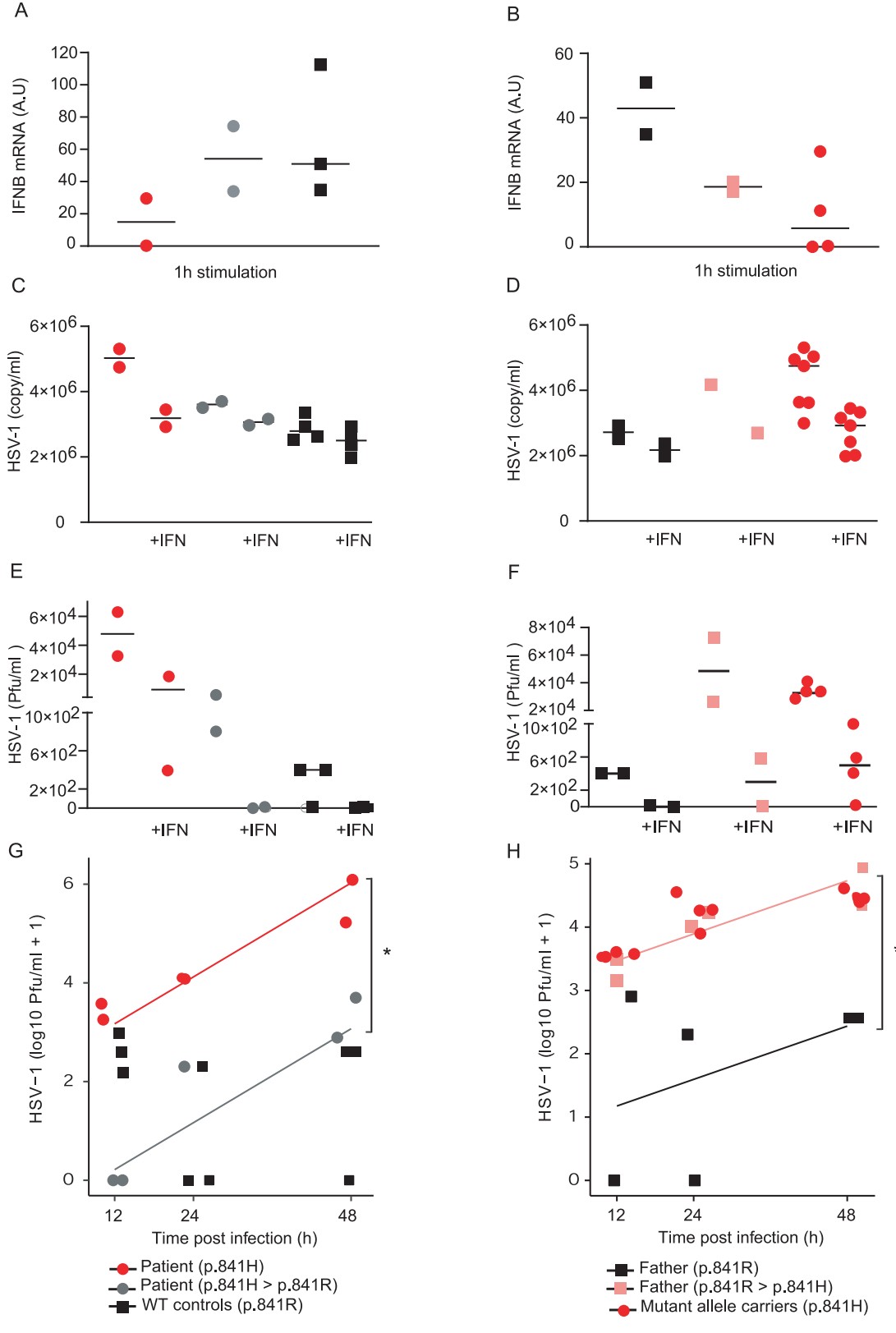

3.30.3[65]). After filtering, the number of genes used for further analysis was 15,306. A quality control analysis was performed through samples correlation/clustering and PCA. Possible confounding factors were considered and removed using the RUVr method from the RUV-seq_1.30.0 R package with $k = 2$.

Differential expression was computed with the R Bioconductor package limma[66] by fitting data to a linear model. P-values were adjusted using the Benjamini-Hochberg (BH) method (false discovery rate (FDR) control). For the DE analysis with the RUV-corrected scores, the two RUVseq factors were directly included in the design as control variables. After testing the DE genes for an FDR < 0.05, the genes were also, in a second step, filtered according to both the FDR (> 0.05) and the fold-change (> 1.5) between "medium" and "stimulation". The log2 fold-change cutoff was set to

**Fig. 8 | Rescue of antiviral immunity in iPSCs-derived neurons. A** Induction of IFNβ mRNA after 1 h of Poly(I:C) stimulation in neurons from WWP2 p.R841H (red, $N = 2$), WWP2 p.H841R (grey, $N = 2$) individuals and WWP2 p.R841 controls (black, $N = 3$). **B** Induction of IFNβ mRNA after 1 h of Poly(I:C) stimulation in neurons from WWP2 p.R841 (black, $N = 2$), WWP2 p.R841H (grey, $N = 2$) individuals and WWP2 p.H841 controls (red, $N = 4$). Results represent the mean ± standard error of 1 experiment among 2. **C** Quantification of the number of viral genome in neurons from WWP2 p.R841H (red, $N = 2$), WWP2 p.H841R (grey, $N = 2$) individuals, and WWP2 p.R841 controls (black, $N = 4$), without or with an IFNα pretreatment (+IFN). **D** Quantification of the number of viral genome in neurons from WWP2 p.R841 (black, $N = 2$), WWP2 p.R841H (grey, $N = 1$) individuals and WWP2 p.H841 controls (black, $N = 7$) without or with an IFNα pretreatment (+IFN). **E** Quantification of the number of plaque-forming units (pfu) by plaque-titration after infection of neurons from WWP2 p.R841H (red, $N = 2$), WWP2 p.H841R (grey, $N = 2$) individuals and WWP2 p.R841 controls (black, $N = 3$) without or with an IFNα2b pretreatment (+IFN). **F** Quantification of the number of plaque-titration after infection of neurons from WWP2 p.R841 (black, $N = 2$), WWP2 p.R841H (grey, $N = 2$) individuals and WWP2 p.H841 controls (black, $N = 4$) without or with an IFNα2b pretreatment (+IFN). **G** Quantification of the number of plaque-forming units (pfu) by plaque-titration at different time after infection of neurons from WWP2 p.R841H (red, $N = 2$), WWP2 p.H841R (grey, $N = 2$) individuals and WWP2 p.R841 controls (black, $N = 3$). A significant effect (*) of the rescue (grey) compared to allele carrier (red) ($p = 2.41e\text{-}5$) was found using a linear regression on $\log10 + 1$ transformed pfu with a time-dependent effect. **H** Quantification of the number of plaque-forming units (pfu) by plaque-titration at different time after infection of neurons from WWP2 p.R841 (black, $N = 2$), WWP2 p.R841H (grey, $N = 2$) individuals and WWP2 p.H841 controls (black, $N = 4$). A significant effect (*) of the mutant induced (pink) compared to control (black) ($p = 2.06e\text{-}3$) was found using a linear regression on $\log10 + 1$ transformed pfu with a time-dependent effect.

0.5849625 so that only genes with an absolute fold change superior to 1.5 were kept.

AP-1 and NF-kB downstream enrichment for the difference in stimulation between wild-type [WT] and mutant [MUT] were computed as follows. A stimulation effect was calculated as log2 fold-change of CPMs between Poly(I:C) stimulation and Medium (ΔStimulation) within each pair of samples. The effect of mutation was then calculated as the difference between stimulations in mutant samples and wild-type samples (ΔΔStimulation = ΔStimulation[Mut] − ΔStimulation[WT]). AP-1 and NF-kB downstream genes were taken from the TNF signaling pathways and NF-kB signaling pathway in KEGG (hsa:04064 and hsa:04668 respectively). Enrichment for ΔΔStimulation in NF-kB-atypical, NF-kB-non-canonical, NF-kB-canonical and AP-1 downstream genes in TNF pathways were computed with Bioconductor R package fgsea.

### Production of recombinant proteins

The WT and mutant WWP2 cDNAs were amplified by PCR from total RNA isolated from PBMCs. Purified products were subcloned into a pGEM®-T Easy vector by T4 DNA ligase (Promega) and sequenced. We generated pcDNA3.1 expression vectors (Thermofisher) encoding the WT and mutant WWP2. Plasmids containing cDNAs were used for the transient transfection of TLR3-expressing human embryonic kidney (HEK) 293 T cells (ATCC® CRL-3216). Briefly, plasmids were mixed with JetPEI (Polyplus transfection, USA), according to the manufacturer's instructions. Cells were cultured for 24 h, 48 h, or 72 h and washed 2 times with phosphate-buffered saline (PBS) before being lysed with a lysis buffer containing Tris 10 mM, pH 8, NaCl 150 mM, NP-40 0.5%, NaF 10 mM, Na-orthovanadate 1 mM and protease inhibitors.

### Expression of recombinant proteins

Cells were harvested with the lysis buffer previously described and centrifuged at $10,000 \times g$ at 4 °C, for 5 min. Protein content was determined by the Biorad Protein assay (Biorad). Proteins (20 µg–40 µg) were then subjected to SDS-PAGE under reducing and then transferred onto a nitrocellulose membrane. The rabbit anti-human WWP2 antibody (1/1000, A302-935, Bethyl Laboratories) and mouse anti-B actin antibody (1/1000, sc-8432, Santa Cruz Biotechnology) were used as primary antibodies and were detected using anti-rabbit or mouse IgG antibodies conjugated to horseradish peroxidase (1/10,000, Amersham Biosciences). Detection was done with the ECL chemiluminescence kit (Pierce) according to the manufacturer's protocol.

### Human inducible pluripotent stem cells (iPSC)

Human iPSCs were generated from the HSE patient, family members as well as healthy controls by reprogramming erythroblasts via ectopic expression of a minimal set of transcription factors as previously described[67]. Briefly, erythroblasts were amplified from PBMCs and nucleofected with episomal plasmids pCXLE-hOCT3/4-shp53-F, pCXLE-hSK, pCXLE-hUL (gifts from Shinya Yamanaka, Addgene #27077, #27080 and #27078)[68], plated on Matrigel-coated plates and cultured in a reprogramming medium (ReproTeSR, STEMCELL Technologies) until iPSC colonies started to appear. Human iPSC clones were then picked and lines expanded using StemMACS iPSC-Brew XF medium (Miltenyi Biotec, Bergisch Gladbach). Human iPSCs were characterized by immunofluorescence for pluripotency (expression of OCT4, TRA-1-60, and SOX2) and differentiation capacity (The three-germ layer differentiation capacity and expression of AFP, Pax6, SMA) as previously described[67] and following standard quality controls for iPSCs. The genomic integrity of iPSCs was assessed by G-banding karyotype. The p.R841H mutation was confirmed by the Sanger sequencing of iPSCs' genomic DNA. Two lines per donor were used in this study. Healthy control iPSC lines ($n = 3$, each from a different individual) were previously described[36] and assessed to be WT for WWP2.

### Inducible pluripotent stem cell culture and differentiation

Human iPSCs were maintained in StemMACS human iPSC Brew medium (Miltenyi) or TeSR-E8 medium (Stemcell Technologies). The differentiation of iPSCs into neuronal precursor cells (NPCs) was promoted as previously described[69]. Subsequent differentiation into astrocytes was performed as detailed elsewhere[69]. Differentiation of NPCs into neurons was conducted by neurogenin 2 overexpression (NGN2). Briefly, NGN2 coding sequence was cloned into the pCW57.1 backbone (gift from David Root (Addgene plasmid # 41393). Human iPSC-derived NPCs were transduced with a VSV-G pseudotyped lentivirus carrying the LV-NGN2 transgene and amplified as previously described[69] in the presence of puromycin (2 µg/ml). Cells were then plated on plates coated with poly-L-ornithine and Laminin in DMEM/F-12 medium supplemented with N-2 and B-27 supplements (1x) (Thermofisher), doxycycline (2 µg/ml) and Laminin (2 µg/ml). Medium was changed every two to three days. At day 8, medium was supplemented with BDNF (10 µg/ml) and GDNF (10 µg/ml). Neurons were used for the experiments after 14 days of differentiation.

### CRISPR-Edited iPSCs

CRISPR Cas9-mediated knockout cells were generated by Synthego Corporation (Redwood City, CA, USA). Briefly, ribonucleoproteins containing the Cas9 protein and synthetic chemically modified sgRNA produced at Synthego were electroporated into the cells along with a single-stranded oligodeoxynucleotide (ssODN) donor using Synthego's optimized protocol. Editing efficiency was assessed upon recovery, 48 h post electroporation. Genomic DNA was extracted from a portion of the cells, PCR amplified, and sequenced using Sanger sequencing. The resulting chromatograms were processed using Synthego Inference of CRISPR edits software (ice.synthego.com). To create monoclonal cell populations, edited cell pools were seeded at 1 cell/well using a single cell printer into 96 or 384 well plates. All wells were imaged every 3 days to ensure expansion from a single-cell clone.

Clonal populations were screened and identified using the PCR-Sanger-ICE genotyping strategy described above. Genomic stability was assessed by KaryoStat+™, G-banding and Short tandem repeat analysis. Pluripotency was verified by PluriTest ™ and immunohistochemical analysis. Cells were tested for bacterial and fungal contamination.

### In vitro ubiquitination assay

Both p.R841 WWP2, p.R841H WWP2, and TRIF were expressed by using the TNT Quick Coupled Transcription Translation Systems kit (Promega), following manufacturer's information. In vitro ubiquitination assay was determined with a ubiquitination kit (Enzo Life Science) according to manufacturer's instructions. Briefly, E1 (100 nM) and UbcH5c (50 μg/ml) as an E2 were added for ubiquitination assays. The reactions were incubated for 90 min at 37 °C and stopped by adding 2x non-reducing gel loading buffer. Ubiquitin-conjugated TRIF was detected by immunoblot with an anti-ubiquitin antibody (1/1000, UBCJ2, ENZ-ABS840, ENZO). The expression levels of WWP2 were also verified by immunoblots with an anti-WWP2 antibody (1/1000, A302-935, Bethyl Laboratories).

### Statistics and reproducibility

No statistical method was used to predetermine sample size. No data were excluded from the analyses. The experiments were not randomized. The Investigators were not blinded to allocation during experiments and outcome assessment.

### Reporting summary

Further information on research design is available in the Nature Portfolio Reporting Summary linked to this article.

## Data availability

The raw data generated in this study are provided in the Supplementary Information/Source Data file. The raw patient's sequences are protected and are not publicly available due to data privacy laws. This data is available from the corresponding author upon request and subject to a data transfer agreement. Source data are provided with this paper.

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

## Acknowledgements

We thank the patient and her family who agreed to participate and made this study possible. We thank Aurélie Guillet, Corine Guyat-Jacques, and Ghislaine Aubel who contributed to the management of the study patient and volunteers. P.Y.B. is supported by the Swiss National Science Foundation (31CA30_196036, 33IC30_179636, and 314730_192616), the Leenaards Foundation, the Santos-Suarez Foundation as well as grants allocated by Carigest. CR is supported by the Swiss National Science Foundation (31CA30_196036 and 31003A_176097). RDP is supported by the Swiss National Science Foundation 320030_179531.

## Author contributions

S.B., M.Q., S.P., C.R., R.D.P., P.Y.B. conceived the study. S.B., M.Q., S.P., A.M., F.S.K., V.Z., N.G., M.J., R.D.P., C.R., P.Y.B. designed the experiments. M.Q., C.R., N.G., and M.J. performed statistical analysis. F.S.K., V.Z. performed molecular modeling. R.C.M., N.F., E.R.P., P.M., F.F., S.A., P.Y.B. performed clinical investigations. S.B., S.P., E.C., M.C. and L.L.C. performed laboratory investigations. E.C., M.C., A.M., P.M., R.B., O.O., L.L.C., F.F., R.D.P., P.Y.B. performed administrative, technical, or material support. S.B., M.Q., C.R. and P.Y.B. wrote the manuscript with input from all authors. C.R., R.D.P., P.Y.B. supervised. P.Y.B. obtained funding for the work.

## Competing interests

The authors declare no competing interests.

## Informed consent

The patient and healthy volunteers included in this study signed an informed consent form for genetic and functional testing, according to protocols approved by the Cantonal Ethics Committee of the state of

Vaud (CER-VD 479/13, CER-VD 2019-02283, CER-VD 2020-01108, CER-VD 2018-01622). Samples were stored within a dedicated biobank fulfilling quality standards according to the Swiss Biobanking Platform criteria ("Vita label", certificate CHUV_2004_3 and CHUV_2103_14).
