## [Peer Review File · Nature Communications]

Herpes simplex encephalitis due to a mutation in an E3 ubiquitin ligaseEditorial Note: Parts of this Peer Review File have been redacted as indicated to remove third-party material where no permission to publish could be obtained.

REVIEWER COMMENTS

Reviewer #1 (Remarks to the Author):

In this paper, Bibert et al describe the investigation of a heterozygous WWP2 missense mutation, in a girl with Herpes simplex encephalitis (HSE). The study is interesting. It is clear that the authors have made great effort in this single patient study, to characterize the impact of the WWP2 mutation both at molecular and cellular virological levels.

However, in its current form, the experimental evidence is still weak in demonstrating the impact of the WWP2 mutation being responsible for an impaired TLR3 signaling and HSV-1 growth control in central nervous cells. I have some suggestions to improve the study:

1. Please provide more specific description of the 'in silico process' that the authors used to select potential HSE relevant gene mutations, leading to their focus on the WWP2 mutation. A Figure or table or tabular presenting the process and selection criteria will be helpful.

2. Other than WWP2, does the patient carry any other mono-allelic or bi-allelic possible gene mutations relevant to her HSE pathogenesis? How those were excluded? In particular, any homozygous, compound-heterozygous, or de novo mutations have been observed to be any potential relevance? Whole exome sequencing of the Trios (patient + 2 parents) will be important in order to assess any other potential gene mutations, prior to focus on WWP2.

3. Figure 2 is not convincing.

- Fig. 2A shows that the R841H mutant consistently display higher protein expression levels than wild-type (WT) WWP2, at all time points tested. Is this reproducible in all experiments? If so, what is the mechanism? A simple qPCR of WWP2 in the same experiment will help to firstly clarify whether the same amount of plasmids were transfected to the cells.

- In Fig. 2B, the fold induction of IFNB by poly(I:C) upon WT WWP2 plasmid transfection was too low (~1.5). The authors should either use HEK293 cells with higher levels of TLR3 expression (plasmid-transfection mediated expression could do the job), or another cell type that has higher endogenous TLR3 activity, e.g. human dermal fibroblasts. Also, cells without exogenous WWP2 expression must be included in these experiments.

4. The authors propose that the R841 WWP2 mutation resulted in impaired TLR3 signaling due to a dysregulation of TRIF ubiquitination upon poly(I:C) stimulation. This is a good hypothesis. They should test this hypothesis experimentally.

5. The data on HSV-1 genome copy numbers in various cell types are overall weak. Although there was a trend of higher HSV-1 levels in the mutant cells, the difference as compared to that of the WT cells were subtle (~ twice of WT cells' levels in general). Moreover, the authors only quantified the HSV-1 genome copy numbers by qPCR, at one time point after viral infection, therefore it is not possible to conclude that there was indeed higher virus replication in the mutant cells. The authors should quantify HSV-1 levels at various time points following infection, with a method that measures the reproductive viral particles, e.g. TCID50 assay.

Some other minus points:

1. The REF#12 is from 2013. There are more recent reviews and papers on this topic. Please update.

2. Which HSV-1 gene was assessed in the HSV-1 genome copy number quantification assay? Please clarify.

3. Fig. 1C, family members carrying the same heterozygous WWP2 mutation should be indicated with a vertical bar.

4. For all applicable figures, please clarify how many times the experiment has been performed. A minimum of 3 experiments is necessary. The current legends are not clear on this point.

Reviewer #2 (Remarks to the Author):

In this manuscript, Bibert et al. reported a clinical case of herpes simplex encephalitis with WWP2 R841H point mutation, which could be a potential gain-of-function mutation contributing to the disease. The Bochud group has a successful record of studying the relationship between SNP and infectious diseases. In this study, they applied exome sequencing and subsequent analysis to identify WWP2 R841H as a key mutation in the 14-year girl patient. This variant was also identified in other family members shown by Sanger sequencing. To support their hypothesis, the authors used HEK293 cells, PBMCs and patient iPSCs derived neuronal cells carrying wt or mutant WWP2 to test their response to poly(I:C) or HSV-1. They concluded that WWP2 R841H mutation might attenuate anti-viral immunity by reducing TLR3 signaling. WWP2 belongs to the NEDD4 family HECT E3 ligases which play critical roles in neuronal development and immune system regulation. A previous genetic study of periventricular nodular heterotopia (PMID: 27694961) has linked HECT E3 mutation to neuronal disease. This study could indicate a potential role of WWP2 in anti-viral immunity and explain the genetics underlying HSE, an infectious disease. In this regard, this study could be of interest to the ubiquitin field as well as immunology field. However, this manuscript does not contain enough evidence to fully support the conclusion the authors are trying to reach, thus publication in Nature Communication is not recommended in its current format.

1. The authors were claiming that R841H variant is a gain-of-function mutation for WWP2, but there is no direct evidence. To support this, the authors should generate recombinant WWP2 proteins with or without the mutation and compare their E3 ligase activities using in vitro ubiquitination assays. It is intriguing how this mutation could affect WWP2's activity. Based on the crystal structure, WWP2 R841 is a residue that is very close to its catalytic cysteine, which might affect the ubiquitin ligase activity of WWP2.
2. Data provided in this manuscript is mainly phenotypical, which lacks mechanistical investigation. Specifically, cells with WWP2 variant showed different TLR3 signaling but the authors did not experimentally depict why. The authors were speculating that it could be TRIF being targeted by WWP2 for degradation that affected TLR3 signaling. Although TRIF has been demonstrated by others to be ubiquitinated and degraded by WWP2, the authors should experimentally test whether it holds true in the context. Substrate specificity of E3 ligases could be cell type specific, thus conclusions from other studies might not be applicable to this one. The authors should carry out a series of experiments to prove that TRIF is substrate of WWP2 in neuronal cells, WWP2 mutation leads to more efficient TRIF degradation, overexpressing TRIF could rescue the phenotype, et al.
3. The authors should provide a more comprehensive analysis of TLR3 signaling pathway instead of merely showing the qPCR results of two downstream target genes. How about the TBK1-IRF3, NF-kB and AP-1 pathway? What about other downstream target genes?
4. It is elegant to use iPSCs from patient, patient's family member, unrelated individuals and differentiate them to neurons to model the infection. However, different individual could also carry very different genetic and epigenetic background that complicates the analysis. To more robustly support the conclusion, the authors should also use a canonical embryonic stem cell line, CRISPR edit the WWP2 gene, carry out the differentiation and the infection analysis.
5. In the experiments using human cell derived samples (PBMC and iPSC), it will be interesting to see which data point corresponds to patient with HSE among the four and which data point corresponds to patient's father among the eight controls. This would help to judge if other possible factor has contributed to the result.

Reviewer #3 (Remarks to the Author):

Here Bibert et al describe the identification of a patient with HSE and heterozygous for a variant in the ubiquitin ligase WWP1.

The authors very convincingly demonstrate impaired/reduced IFN and ISG responses in patient PBMCs and iPSCs and also demonstrate a phenotype in neurons and stem neuronal progenitor cells (but not astrocytes) which can be reversed by expression of the WT and also by IFN α pretreatment.

Based on previous papers they suggest the mechanism to be WWP1 mediated degradation of TRIF, thereby disturbing TLR3 signaling and IFN responses

Overall the work is impressive and interesting. However major improvements to further unravel the molecular mechanism are required

1. Genetics: please also state the CADD score and MSC
2. Generally PBMCs are not considered permissive for HSV infection. Could the authors possibly obtain fibros from the patient and do infection experiments in those? Alternatively mainly base conclusion on infection in neurons and other permissive cells
3. A major point concerns the molecular mechanism which is not well characterized. If indeed WWP1 lead to increased ubiquitination and proteasome-mediated degradation of TRIF this should be examined, including
 - i) Ubiquitination activity of WWP1 on relevant substrate
 - ii) Accelerated TRIF degradation in patient cells, i.e poly(IC)-induced and/or HSV induced decrease in TRIF protein levels in patient cells compared to controls
 - iii) Reversion of this effect by blocking of the proteasome with proteasome inhibitors like bortezomib
 - iv) Show one or more signaling molecules in the IFN induction pathway that are hypo-ubiquitinated or degraded?
4. The authors could make a structure of WWP1 and show how the variant in the specific domain is predicted to interfere with structure or activity of the molecule
- 5) Is the defect specific for the TLR3 pathway ? Are other pathway such as cGAS (dsDNA), RIG-I (transfected poly(IC) and TLR9 (CpG DNA) intact?
- 6) iPSC-derived microglia should be tested as well. Neurons produce limited IFN during HSV infection so microglia may play an essential role and the magnitude of the latter response is far greater?

Specific comments to figures

Show asterisk as per significance level in all figures

Lausanne, December 22th, 2023

Re: NCOMMS-22-48925A, Manuscript entitled “ Herpes Simplex Encephalitis due to Gain of Function Mutation in an E3 Ubiquitin Ligase”.

Response to reviewers

Reviewer #1:

In this paper, Bibert et al describe the investigation of a heterozygous WWP2 missense mutation, in a girl with Herpes simplex encephalitis (HSE). The study is interesting. It is clear that the authors have made great effort in this single patient study, to characterize the impact of the WWP2 mutation both at molecular and cellular virological levels.

However, in its current form, the experimental evidence is still weak in demonstrating the impact of the WWP2 mutation being responsible for an impaired TLR3 signaling and HSV-1 growth control in central nervous cells. I have some suggestions to improve the study:

1. Please provide more specific description of the ‘in silico process’ that the authors used to select potential HSE relevant gene mutations, leading to their focus on the WWP2 mutation. A Figure or table or tabular presenting the process and selection criteria will be helpful.

RESPONSE. We thank the reviewer for the comment. The selection process of relevant variants is now described in the text and in the **Figure 1 (see new panel A)**. Please refer to point 2 for a more complete description.

2. Other than WWP2, does the patient carry any other mono-allelic or bi-allelic possible gene mutations relevant to her HSE pathogenesis? How those were excluded? In particular, any homozygous, compound-heterozygous, or de novo mutations have been observed to be any potential relevance? Whole exome sequencing of the Trios (patient + 2 parents) will be important in order to assess any other potential gene mutations, prior to focus on WWP2.

RESPONSE. Indeed, to address this important point, we sequenced the proband's parents DNA and performed a **whole exome sequence (WES)-based trio analysis**. Analyses were performed for all possible inheritance modes, namely the homozygous, compound heterozygous, *de novo*, or heterozygous states, as detailed in **Supplementary Table 1**.

All potentially relevant variants were selected based on their quality, frequency in population, impact at the protein level, amino acid conservation as well as deleteriousness predicted by *in silico* tools (CADD and MutScore). The previous links with immunity and TLR3 pathway was investigated by looking at diseases known to be caused by candidate genes as well as phenotypes in knock-out mice.

For heterozygous variants, including *de novo* variant, the DOMINO score (Quinodoz, M. *et al. Am J Hum Genet* **101**, 623-629, 2017) was used to predict whether corresponding genes were likely linked to autosomal dominant phenotypes or not. Heterozygous variants in genes with low DOMINO scores were therefore discarded.

Specifically, among 155 identified variants, all were at the heterozygous state, including 5 heterozygous compound variants. Among the 5 compound variants, 3 were located within the FLNB gene (NM_001457.4:c.2237A>C, c.4186G>A, and c.5327T>C), among which two were predicted deleterious. An association with HSE was unlikely, as FLNB can carry variants associated with phenotypes affecting the skeleton at an autosomal dominant state. The two other compound variants were located within the ZFYVE26 gene (NM_015346.4:c.3881A>G and c.961A>C), but the first one affected a non-conserved amino acid and the gene was already associated with another phenotype, i.e. spastic paraplegia. Among the 150 remaining heterozygous variants, 2 were *de novo* variants located in RBM13 (NM_002870.4:c.429T>G) and in CNTNAP2 (NM_014141.6:c.3427G>C), with low DOMINO scores, suggesting that they are not likely to induce a dominant phenotype. Among the 148 remaining heterozygous variants, 18 had a high DOMINO score, but only 2 of them were predicted to be deleterious by Combined Annotation Dependent Depletion (CADD) and MutScore. The first missense variant was identified in VANG2 (NM_020335.3:c.931G>A), a gene that can carry variants causing autosomal dominant neural tube defects. The second variant was detected in the WW Domain containing E3 Ubiquitin protein ligase 2 gene (WWP2) which was not previously associated with a genetic disorder. Since WWP2 was shown to downregulate the TLR3/IFN pathway, which is essential in the immune response against HSV-1, it was considered our primary target.

It is important to note that Herpes Simplex encephalitis due to genetic defects in the TLR3 pathway is known to display incomplete penetrance, e.g. not all carriers of the mutations will develop the disease. This is why in our analysis, we retained heterozygous variants with non-zero frequency in controls (gnomAD database). For the analysis of causative variants in fully penetrant disease, heterozygous variants with any frequency in controls would normally be discarded.

3. Figure 2 is not convincing.

- Fig. 2A shows that the R841H mutant consistently display higher protein expression levels than wild-type (WT) WWP2, at all time points tested. Is this reproducible in all experiments? If so, what is the mechanism? A simple qPCR of WWP2 in the same experiment will help to firstly clarify whether the same amount of plasmids were transfected to the cells.

RESPONSE: we thank the reviewer for this remark; indeed, the slightly different expression of the p.R841H WWP2 was due to differential amounts of the transfected plasmid, as confirmed by qPCR control (see below).

The expression of endogenous p.R841 WWP2 (1), empty vector (2), p.R841 WWP2 (3) and p.R841H WWP2 (4) proteins was determined by SDS-PAGE/immunoblotting under reducing conditions in cell lysates of TLR3-expressing 293T cells after transfection. The mRNA amount of transfected WWP2 was verified by q-PCR. A.U. stands for arbitrary units. A and B represent 2 independent experiments.

- In Fig. 2B, the fold induction of IFNB by poly(I:C) upon WT WWP2 plasmid transfection was too low (~1.5). The authors should either use HEK293 cells with higher levels of TLR3 expression (plasmid-transfection mediated expression could do the job), or another cell type that has higher endogenous TLR3 activity, e.g. human dermal fibroblasts. Also, cells without exogenous WWP2 expression must be included in these experiments.

RESPONSE: we agree that the experiment was not optimal to characterize the difference in IFN induction between p.R841 WWP2 and p.R841H WWP2. We propose to discard this panel, as the issue is clearly addressed in subsequent experiments with most appropriate settings (PBMCs and iPSc-derived cells).

4. The authors propose that the R841 WWP2 mutation resulted in impaired TLR3

signaling due to a dysregulation of TRIF ubiquitination upon poly(I:C) stimulation. This is a good hypothesis. They should test this hypothesis experimentally.

RESPONSE: We agree with the reviewer that assessing the mutant-associated dysregulation of TRIF ubiquitination is a relevant addition. Both p.R841 WWP2, p.R841H WWP2 and TRIF were expressed by using the TNT Quick Coupled Transcription Translation Systems kit (Promega), following manufacturer's information. *In vitro* ubiquitination assay was determined with an ubiquitination kit (Enzo Life Science) according to manufacturer's instructions. Briefly, E1 and Ubch5c as an E2 were added for ubiquitination assays. Ubiquitin-conjugated TRIF was detected by immunoblot with an anti-ubiquitin antibody. The expression levels of WWP2 were also verified by immunoblots with an anti-WWP2 antibody. Altogether, this assay suggests that both p.R841 WWP2 and p.R841H WWP2 were able to directly ubiquitinate TRIF but with a higher extent for p.R841H WWP2. This observation has been added as a **new Figure 8**.

The data on HSV-1 genome copy numbers in various cell types are overall weak. Although there was a trend of higher HSV-1 levels in the mutant cells, the difference as compared to that of the WT cells were subtle (~ twice of WT cells' levels in general). Moreover, the authors only quantified the HSV-1 genome copy numbers by qPCR, at one time point after viral infection, therefore it is not possible to conclude that there was indeed higher virus replication in the mutant cells. The authors should quantify HSV-1 levels at various time points following infection, with a method that measures the reproductive viral particles, e.g. TCID50 assay.

RESPONSE: In line with the reviewer's suggestion, we measured the infectious titers of HSV-1 in neurons at different time points (12, 24, 48 hours after viral infection) by plaque assays. Briefly, the infectivity was evaluated by the cytopathic effect (in plaque forming units) in Vero cells. Neurons from p.R841H carriers exhibited higher viral replication rates compared to controls at all these time-points, thereby confirming the results obtained by

qPCR. These observations have been added as a new Figure 7 and new panels (E-H) in Figure 9.

Some other minus points:

1. The REF#12 is from 2013. There are more recent reviews and papers on this topic. Please update.

RESPONSE: more recent reviews have been cited instead.

2. Which HSV-1 gene was assessed in the HSV-1 genome copy number quantification assay? Please clarify.

RESPONSE: We used a PCR test implemented in the microbiology laboratory of our Institution, which is routinely used to test clinical samples and targets viral glycoprotein B (gB) (Greub, G., Sahli, R., Brouillet, R. & Jaton, K. *Future Microbiol* **11**, 403-25, 2016).

3. Fig. 1C, family members carrying the same heterozygous WWP2 mutation should be indicated with a vertical bar.

RESPONSE: we thank the reviewer for this remark; indeed, in accordance with the standard representation, a point was added.

4. For all applicable figures, please clarify how many times the experiment has been performed. A minimum of 3 experiments is necessary. The current legends are not clear on this point.

RESPONSE: The number of experiments has been reported in Figure legends.

Reviewer #2

In this manuscript, Bibert et al. reported a clinical case of herpes simplex encephalitis with WWP2 R841H point mutation, which could be a potential gain-of-function mutation contributing to the disease. The Bochud group has a successful record of studying the relationship between SNP and infectious diseases. In this study, they applied exome sequencing and subsequent analysis to identify WWP2 R841H as a key mutation in the 14-year girl patient. This variant was also identified in other family members shown by Sanger sequencing. To support their hypothesis, the authors used HEK293 cells, PBMCs and patient iPSCs derived neuronal cells carrying wt or mutant WWP2 to test their response to poly(I:C) or HSV-1. They concluded that WWP2 R841H mutation might attenuate anti-viral immunity by reducing TLR3 signaling. WWP2 belongs to the NEDD4 family HECT E3 ligases which play critical roles in neuronal development and immune system regulation. A previous genetic study of periventricular nodular heterotopia (PMID: 27694961) has linked HECT E3 mutation to neuronal disease. This study could indicate a potential role of WWP2 in anti-viral immunity and explain the genetics underlying HSE, an infectious disease. In this regard, this study could be of interest to the ubiquitin field as well as immunology field. However, this manuscript does not contain enough evidence to fully support the conclusion the authors are trying to reach, thus publication in Nature Communication is not recommended in its current format.

1. The authors were claiming that R841H variant is a gain-of-function mutation for

WWP2, but there is no direct evidence. To support this, the authors should generate recombinant WWP2 proteins with or without the mutation and compare their E3 ligase activities using *in vitro* ubiquitination assays. It is intriguing how this mutation could affect WWP2's activity. Based on the crystal structure, WWP2 R841 is a residue that is very close to its catalytic cysteine, which might affect the ubiquitin ligase activity of WWP2.

RESPONSE: We performed molecular modeling analyses of the impact of WWP2 mutation and estimated the energetic impact of the variant compared to the WT structure. These data strongly suggest that the histidine residue could compromise the structural integrity of the catalytic region of the protein by impacting the WT interaction network (Figure 2 A & B, supplementary Figure 3, supplementary Table 2). This is confirmed in the calculation of the change in the folding free energy, which is significantly higher in the mutant compared to the WT. Altogether, the mutation should lead to a more flexible region and facilitate the accessibility of the ubiquitination site at Cys838.

2. Data provided in this manuscript is mainly phenotypical, which lacks mechanistical investigation. Specifically, cells with WWP2 variant showed different TLR3 signaling but the authors did not experimentally depict why. The authors were speculating that it could be TRIF being targeted by WWP2 for degradation that affected TLR3 signaling. Although TRIF has been demonstrated by others to be ubiquitinated and degraded by WWP2, the authors should experimentally test whether it holds true in the context. Substrate specificity of E3 ligases could be cell type specific, thus conclusions from other studies might not be applicable to this one. The authors should carry out a series of experiments to prove that TRIF is substrate of WWP2 in neuronal cells, WWP2 mutation leads to more efficient TRIF degradation, overexpressing TRIF could rescue the phenotype, et al.

RESPONSE: We agree with the reviewer that assessing the mutant-associated dysregulation of TRIF ubiquitination is an important issue. By using an *in vitro* ubiquitination assay, we now show that WWP2 p.R841H induces increased TRIF ubiquitination compared to WWP2 p.R841. This observation has been added as new Figure 8. The sole production of a new set of neurons requires at least 1 month, thereby limiting our ability to set up and repeat novel experiments. Yet, we understand the reviewers' point and extensively modified the phrasing to account for these limitations. See also response to reviewer 1, point 4.

3. The authors should provide a more comprehensive analysis of TLR3 signaling pathway instead of merely showing the qPCR results of two downstream target genes. How about the TBK1-IRF3 (Rantes?), NF-kB and AP-1 pathway? What about other downstream target genes?

RESPONSE: to have an overall transcriptomic profile and to identify more differentially expressed genes, we have performed RNA sequencing from neurons from 3 WWP2 p.R841H carriers and 3 controls, under basal conditions and after 3h poly(I:C) stimulation. We found that the downstream pathways of NF-kB and AP-1 were significantly enriched for an inhibition of the stimulation in WWP2 p.R841H compared to control (i.e. $\log_2[\text{poly(I:C)} - \text{basal}]$ effect in WWP2 p.R841H is consistently $<$ than control). Given the limited number of samples, our transcriptomic approach was not deep enough to accurately capture direct downstream target of TBK1-IRF3 like IFN-B expression; however, reduced expression of IFNB and ISG had been previously assessed by RT-PCR. These data have been added to the results section and now refer to Supplementary Tables 3 and 4, as well as supplementary Figures 4 and 5 (KEGG pathways downstream of NF-kB and AP-1).

Supplementary Table 4

Pathway	FDR adjusted-pvalue	Normalized Enrichment Score
NF-kB atypical	4.02e-01	1.09
NF-kB non-canonical	4.02e-01	-1.10
NF-kB canonical	3.02e-02	-1.65
TNF: AP-1 Downstream	3.55e-05	-2.07

4. It is elegant to use iPSCs from patient, patient's family member, unrelated individuals and differentiate them to neurons to model the infection. However, different individual could also carry very different genetic and epigenetic background that complicates the analysis. To more robustly support the conclusion, the authors should also use a canonical embryonic stem cell line, CRISPR edit the WWP2 gene, carry out the differentiation and the infection analysis.

RESPONSE: It is correct that the 4 individuals carrying the WWP2 R841H mutation are related, and that the observed phenotype may, in principle, not only result from this mutation, but also from other co-inherited genetic and/or epigenetic factors. However, the dysfunctional phenotype was clearly observed in R841H CRISP-Cas9-modified cells from the father, who is not related to the mother. Furthermore, the dysfunctional phenotype was clearly observed in cells from the mother and her 3 children, and it is unlikely that the mother would have transmitted such co-inherited factors to all 3 children.

In addition, the use of canonical ESC line is limited by several factors. They have a genetic background by themselves; they are not considered "canonical" because they represent some standard genome, but because they are broadly available; they underwent many replications circles that increase the risk that they are carrying genetic abnormalities. Strikingly, many mutations observed in HSCs are related to the DNA synthesis and replication cycle. Because intrinsic IFN pathways play a role in controlling DNA replication, these canonical HSC lines may carry mutations in pathways related to the one we are studying thus would not be the most suitable.

5. In the experiments using human cell derived samples (PBMC and iPSC), it will be interesting to see which data point corresponds to patient with HSE among the four and which data point corresponds to patient's father among the eight controls. This would help to judge if other possible factor has contributed to the result.

RESPONSE: we agree with the reviewer and we have highlighted data points corresponding to patient with HSE and to her father (Figures 3, 6, 7),

Reviewer #3

Here Bibert et al describe the identification of a patient with HSE and heterozygous for a variant in the ubiquitin ligase WWP1. The authors very convincingly demonstrate

impaired/reduced IFN and ISG responses in patient PBMCs and iPSCs and also demonstrate a phenotype in neurons and stem neuronal progenitor cells (but not astrocytes) which can be reversed by expression of the WT and also by IFN α pretreatment. Based on previous papers they suggest the mechanism to be WWP1 mediated degradation of TRIF, thereby disturbing TLR3 signaling and IFN responses. Overall the work is impressive and interesting. However major improvements to further unravel the molecular mechanism are required.

1. Genetics: please also state the CADD score and MSC

RESPONSE: this is a good suggestion. We performed sequencing in patients' parents and trio analyses. All potentially relevant variants heterozygous, *de novo* heterozygous, compound heterozygous – none were homozygous) were evaluated (see also response to point 1 and 2 by reviewer 1). The selection process of relevant variants is now described in the text and in Figure 1 (see new panel A). We also added the CADD score. Since the MSC score server was temporarily not available, we used the MutScore instead, which is a recently developed missense deleteriousness predictor taking into account multiple features (Quinodoz, M et al. Am J Hum Genet 109, 457-470, 2022, Supplementary Table 1).

2. Generally PBMCs are not considered permissive for HSV infection. Could the authors possibly obtain fibros from the patient and do infection experiments in those? Alternatively mainly base conclusion on infection in neurons and other permissive cells

RESPONSE: We agree that the use of neuronal cells is the most relevant model to assess susceptibility to HSE. In our paper, PMCs have been used as one approach among other, which all lead to the same observations. Yet, investigators reported that PBMCs (together with human monocytes and macrophages) can produce infectious HSV-1 virions, although are less permissive to viral replication than fibroblasts (Iannello, A. et al. Viral Immunol 24, 11-26, 2011, Larcher, C. et al. J Invest Dermatol 116, 150-6, 2001). Two sentences have been removed from the discussion to down tone conclusions from these cells.

3. A major point concerns the molecular mechanism which is not well characterized. If indeed WWP1 lead to increased ubiquitination and proteasome-mediated degradation of TRIF this should be examined, including

- i) Ubiquitination activity of WWP1 on relevant substrate
- ii) Accelerated TRIF degradation in patient cells, i.e poly(IC)-induced and/or HSV induced decrease in TRIF protein levels in patient cells compared to controls
- iii) Reversion of this effect by blocking of the proteasome with proteasome inhibitors like bortezomib
- iv) Show one or more signaling molecules in the IFN induction pathway that are hypo-ubiquitinated or degraded?

RESPONSE: We agree with the reviewer that assessing the mutant-associated dysregulation of TRIF ubiquitination is a quite relevant addition. By using an *in vitro* ubiquitination assay, we now show that WWP2 p.R841H induces increased TRIF ubiquitination compared to WWP2 p.R841. This observation has been added as a new Figure 8.

Please see also response to reviewer 1 (point 4), and reviewer 2 (point 2), in which we explain the time constraints in performing further experiments, particularly in neurons.

The text has been extensively modified to clearly express study's limitations. We believe that the new experiment in Figure 8 with all the other data provided in this paper provide reasonable evidence to support the hypothesis by which p.R841H induces increased TRIF ubiquitination compared to WWP2 p.R841.

4. The authors could make a structure of WWP1 and show how the variant in the specific domain is predicted to interfere with structure or activity of the molecule

RESPONSE: we agree that modeling of the WT and mutant proteins are important to functionally characterize the variant. Indeed, we performed molecular modeling analyses of the impact of WWP2 mutation and estimated the energetic impact of the variant compared to the WT structure network (Figure 2 A & B, supplementary Figure 3, supplementary Table 2). Altogether, the mutation should lead to a more flexible region and facilitate the accessibility of the ubiquitination site at Cys838 (see detailed response to reviewer 1, point 1).

5) Is the defect specific for the TLR3 pathway ? Are other pathway such as cGAS (dsDNA), RIG-I (tranfected poly(IC) and TLR9 (CpG DNA) intact?

RESPONSE: As shown in Figure 5D, the expression of IFNB was not reduced in p.R841H carriers compared to WT controls when iPSCs-derived cells were stimulated with LPS (TLR4 agonist), CpG (TRL9 agonist) and R848 (TLR7/8 agonists), suggesting that reduced IFNB in p.R841H carriers was specific to the TLR3/IFN axis.

6) iPSC-derived microglia should be tested as well. Neurons produce limited IFN during HSV infection so microglia may play an essential role and the magnitude of the latter response is far greater?

RESPONSE: Brain microglia was shown to protect neurons from HSV-1 infection by producing pro-inflammatory chemokines (CXCL10, CCL3, CCL4 and CCL2), and cytokines (CCL5, TNF, IL-6, IL-8 and IL-1B) in a TLR3-dependent manner (Lokensgard, J.R. *et al. J Neurovirol* 7, 208-19 2001). We agree with the reviewer that it would be interesting to study the impact of the mutation on the microglia function. We have tried to generate iPSC-derived microglia but reproducing published studies in this field is challenging, takes time and would delay too much our original findings about the WWP2 R841H mutation. Taking into account that the major hypothesis in the field regarding genetic susceptibility to HSV encephalitis involves intrinsic neuronal defects, that our data support this hypothesis, we respectfully think that it is out of the scope of the study but a very good idea for a follow up study.

Show asterisk as per significance level in all figures

RESPONSE: Asterisk was used in all figures.

REVIEWER COMMENTS

Reviewer #1 (Remarks to the Author):

The authors have properly addressed my previous comments and those from the other reviewers.

A few suggestions to further improve the paper:

1. On page 4, lines 114-115, when talking about data from the gnomAD database, please specify the version of gnomAD used. Also, I would remove the word 'normal' from 'normal controls'.

2. On page 5, lines 125, 127, please correct the 'error, reference source not found'.

3. On page 8, the 'TRIF ubiquitination is markedly increased...' section can be combined to the earlier 'Transient gene expression' section. Together they will make a section on the 'molecular characterization by over-expression'.

4. The data presented in the 'Impaired PBMC responses to poly(I:C) and HSV-1' part is overall weak. The cellular phenotype shown in the related figures are very mild. Also, responses to poly(I:C) stimulation in PBMCs are not TLR3-specific (please check previous publications on the topic). I would suggest to move this section to supplementary, or to clearly discuss the limitation of these data as related to 'TLR3'.

5. In the Abstract, please remove this part 'in peripheral blood mononuclear cells and', from the sentence describing that 'the pR841H variant impaired TLR3 mediated signaling in.....cells' from lines 49 to 52.

Reviewer #2 (Remarks to the Author):

In this revised manuscript, the authors performed molecular modeling and in vitro ubiquitination assay to address the biochemical effect of WWP2 mutation. However, the data provided is still very weak to support the KEY conclusion that R841H is a gain-of-function mutation. The current manuscript also suffers greatly from lack of mechanism investigation.

The molecular modeling analysis of the mutation focused on the free energy change and the interaction network within active site of the HECT domain alone. The modeling indicates that R841H mutation could disrupt the interaction and structural integrity as well as increase the free energy, but it does not at all explain or support the activating effect of R841H mutation. Such structural or energetic change could either increase or decrease enzyme activity of HECT E3 ligase. To more convincingly explain the mutation effect, the authors should have a more careful analysis of the modeling. Firstly, WWP2 HECT domain adopts auto-inhibitory conformation, does this mutation activate WWP2 by releasing autoinhibition? The authors should do the modeling of the autoinhibitory structure of WWP2. Secondly, from the perspective of a HECT enzyme mechanism, HECT catalyze ubiquitination process in a multi-step fashion. HECT interaction with E2~Ub, and then E2~Ub transfers Ub to the catalytic Cysteine of HECT, HECT~Ub undergoes a conformational change to transfer Ub to the bound substrate. There are crystal structures for almost every single one of these steps. How does the mutation perturb the machinery in such context? Without careful analysis, the molecular modeling that the authors did has very limited strength to support their statement about WWP2 R841H enzyme activity.

The other major problem is that the in vitro ubiquitination assay data is also pretty weak to support the conclusion. Firstly, the detailed description of the experimental condition is not specified, what are the concentrations of Ub/E1/E2/E3/substrate used in the reaction, and how long was the reaction? Secondly, we usually assess the ubiquitination reaction based on decrease of unmodified substrate protein and appearance of ubiquitinated substrate bands/smear at higher molecular weight. The in vitro ubiquitination assay result presented heavily rely on the blot using antibody against ubiquitin. This could be problematic because the ubiquitin signal does not directly represent TRIF ubiquitination, although the author set up the control with WWP2 without TRIF

which showed no ubiquitination signal. This is also very strange, because we and other groups have done large amount of in vitro ubiquitination assays of WWP2, and we always see the autoubiquitination of WWP2 without substrate, but the authors did not detect any autoubiquitination signal. Thirdly, the change in ubiquitination signal is very subtotal based on the shown blot.

Moreover, there is a huge gap between the activity change of WWP2 R841H and the biological consequences. If it is true that R841H mutation promotes TRIF ubiquitination, what are the biological effects? Does this mutation promote TRIF ubiquitination in the cell? What type of ubiquitination? Does ubiquitination leads to degradation or other effects? Why the enhanced ubiquitination leads to change in viral responses?

Reviewer #3 (Remarks to the Author):

The authors made some improvements as suggested by reviewers's

They should further build the case in regards to

1) molecular mechanism

2) reconstitution experiment (no statistics in Fig 9)

3) demonstration of changes in ubiquitination in patients cells and/or cells expressing the mutant WWP2 protein (Figure 8 is insufficient)

Point per point responses to Reviewers

REVIEWER COMMENTS

Reviewer #1 (Remarks to the Author):

The authors have properly addressed my previous comments and those from the other reviewers.

Response: we thank the reviewer for the useful comments which we believe contributed to improve the manuscript.

A few suggestions to further improve the paper:

1. On page 4, lines 114-115, when talking about data from the gnomAD database, please specify the version of gnomAD used. Also, I would remove the word 'normal' from 'normal controls'.

Response: the version of gnomAD (v2.1.1) has been added. The term "normal controls" has been changed to "controls".

2. On page 5, lines 125, 127, please correct the 'error, reference source not found'.

Response: the problem with the "Error, reference no found" has been removed and replaced by Figure 2A and Figure 2B.

3. On page 8, the 'TRIF ubiquitination is markedly increased...' section can be combined to the earlier 'Transient gene expression' section. Together they will make a section on the 'molecular characterization by over-expression'.

Response: we thank the reviewer for this suggestion, though we would prefer to stick with the current sequence of paragraphs with progression from biochemical to functional and mechanistic characterization.

4. The data presented in the 'Impaired PBMC responses to poly(I:C) and HSV-1' part is overall weak. The cellular phenotype shown in the related figures are very mild. Also, responses to poly(I:C) stimulation in PBMCs are not TLR3-specific (please check previous publications on the topic). I would suggest to move this section to supplementary, or to clearly discuss the limitation of these data as related to 'TLR3'.

Response: we have moved Figure 3 to the "supplementary Figures". In addition, a sentence stating the limitation of poly(I:C) stimulation in PBMCs has been added in the result section and the paragraph on PBMCs was erased from the discussion.

5. In the Abstract, please remove this part 'in peripheral blood mononuclear cells and', from the sentence describing that 'the pR841H variant impaired TLR3 mediated signaling in.....cells' from lines 49 to 52.

Response: reference to PBMCs has been removed from the abstract.

Reviewer #2 (Remarks to the Author):

In this revised manuscript, the authors performed molecular modeling and in vitro ubiquitination assay to address the biochemical effect of WWP2 mutation. However, the data provided is still very weak to support the KEY conclusion that R841H is a gain-of-function mutation. The current manuscript also suffers greatly from lack of mechanism investigation.

Response: we agree with the reviewer that it would be very interesting to conduct molecular dynamics simulations to study the mutation's impact on the region's conformation and additional mechanistic investigations. This would however constitute a standalone project, possibly contributing to the discovery of other key residues involved in WWP2 activity.

The molecular modeling analysis of the mutation focused on the free energy change and the interaction network within active site of the HECT domain alone. The modeling indicates that R841H mutation could disrupt the interaction and structural integrity as well as increase the free energy, but it does not at all explain or support the activating effect of R841H mutation. Such structural or energetic change could either increase or decrease enzyme activity of HECT E3 ligase. To more convincingly explain the mutation effect, the authors should have a more careful analysis of the modeling. Firstly, WWP2 HECT domain adopts auto-inhibitory conformation, does this mutation activate WWP2 by releasing autoinhibition? The authors should do the modeling of the autoinhibitory structure of WWP2.

Response: We agree with the reviewer that WWP2 can adopt an auto-inhibition conformation, similar to WWP1. In Wang et al.'s study (PMID: 31320636), on the multi-lock inhibitory mechanism of the HECT family E3 ligases, it is mentioned that the multi-lock auto-inhibition mode, resulting from the binding of the WW2, L, and WW3-4 domains to the catalytic HECT domain, maintains WWP1 in a fully inactive state. Based on the structural analysis of the published structures (PDB ids: 6j1x, 6j1y), we observe that the region containing the position of interest (WWP1 p.Arg893) within the HECT domain is not at the interface with any other domains. The primary distinction is that the loop (residues 603 to 609) remains unresolved in the fully inactivated structure (PDB id 6j1x). Similarly, we anticipate a similar behaviour for WWP2. (**Figure 1**).

As we presented in the manuscript, WWP1 and WWP2 are two very similar proteins with share a percentage of sequence identity of 83.58% and 63% for the HECT domain and the whole sequence, respectively. The mutation is predicted to not impact the regions involved in the auto-inhibition conformation but rather directly affect the catalytic region of the HECT domain.

Secondly, from the perspective of a HECT enzyme mechanism, HECT catalyze ubiquitination process in a multi-step fashion. HECT interaction with E2~Ub, and then E2~Ub transfers Ub to the catalytic Cysteine of HECT, HECT~Ub undergoes a conformational change to transfer Ub to the bound substrate. There are crystal structures for almost every single one of these steps. How does the mutation perturb the machinery in such context?

Response: One limitation is the small number of WWP2 and HECT domain experimental structures available. The five WWP2 ones used in this study present an important global conformational similarity, with an average RMSD (root mean squared deviation) of 0.65 Å, which is very low and not significant (this difference is below the typical 2 Å uncertainty of X-ray structures, **Table 1**).

RMSD vs. 4y07.A (Å)	
5tjq.A	0.462
5tj7.A	0.793
5tj8.A	0.623
6j1z.A	0.713

Table 1. RMSD values calculated using the chain A of the 4y07 structure and the 4 other structures (Å).

Indeed, many experimental structures involving the different ubiquitination steps exist for HECT family E3 ligases. Most of them share lower sequence identity with WWP2. We did the analysis on some of them, involving ITCH, NEDD4 and NEDD4L (**Table 2**), which share a sequence identity of 78.51%, 55.22%, 54.63%, respectively. Performing the analysis of a larger scale of proteins and organisms could be interesting as a standalone project.

Protein	PDB ID	Position of the Arg
ITCH	5c7m[11]	874
ITCH	4be8[12]	874
NEDD4	2xbb[13]	870
NEDD4	5c7j[11]	870
NEDD4	2xbf[13]	870
NEDD4	4bbn[12]	870
NEDD4	5c7j[11]	870
NEDD4	5c91[14]	870
NEDD4L	3jw0[15]	925
NEDD4L	3jvz[15]	925
NEDD4L	2oni	925
NEDD4L	5hpk[11]	925
NEDD4L	3tug	925
SMURF2	1zvd[16]	719
WWP1	5hpt[11]	893
WWP1	1nd7[17]	893
WWP1	5hps[11]	893
WWP1	6j1y[10]	893
WWP2	4y07[7]	841
WWP2	5tj7[9]	841
WWP2	5tj8[9]	841
WWP2	5tjq[9]	841
WWP2	6j1z[10]	841

Table 2. Experimental structures fetched.

For proteins other than WWP1 and WWP2, the analyzed structures indicate that even when the arginine is in the proximity of a binding partner, the conformation of interest, including the environment of the arginine, remains similar to the apo structures of WWP1 and WWP2. This suggests that its contribution is important to maintain the rigidity of the region. For example, in the structure with the PDB id 3jvz, crystallized in the presence of NEDD4L, Ubiquitin, and UBE2D2, we observe that the conformation of the region of interest is almost identical between the structures with PDB ids 3jvz (NEDD4L) and 4y07 (WWP2), despite a sequence identity of 54.53% (**Figure 2**).

Based on the analyses of various structures, we observed that energy fluctuations tend to destabilize the region. The hypothesis is that this effect makes Cys838 more accessible, as the energy cost required to render it accessible is lower.

Without careful analysis, the molecular modeling that the authors did has very limited strength to support their statement about WWP2 R841H enzyme activity.

Response: We understand the point of view of the reviewer. However, we believe that our analysis provides good support to make the hypothesis that, since the structural conformation where the Arg841 is located is highly conserved and rigid, introducing more flexibility could potentially make the Cys838, key residue in the ubiquitination process, more accessible. We agree that additional work, going beyond this structural analysis, using MD simulations or normal mode analysis for instance, could potentially also decipher the exact effect of the mutation on the final structure. However, as mentioned above, although it is scientifically interesting, we consider that the exact end-point structure after mutation is not necessary to understand and predict that the mutation will have a pathogenic effect – the fact that the wild-type structure is radically perturbed, as we demonstrated, seems reasonable enough to make this conclusion. Further investigations *in silico* could make the object of a separate article focused primarily on molecular modelling.

The other major problem is that the *in vitro* ubiquitination assay data is also pretty weak to support the conclusion. Firstly, the detailed description of the experimental condition is not specified, what are the concentrations of Ub/E1/E2/E3/substrate used in the reaction, and how long was the reaction?

Response: Experimental conditions were detailed in the “methods” section.

Secondly, we usually assess the ubiquitination reaction based on decrease of unmodified substrate protein and appearance of ubiquitinated substrate bands/smear at higher molecular weight. The *in vitro* ubiquitination assay result presented heavily rely on the blot using antibody against ubiquitin. This could be problematic because the ubiquitin signal does not directly represent TRIF ubiquitination, although the author set up the control with WWP2 without TRIF which showed no ubiquitination signal. This is also very strange, because we and other groups have done large amount of *in vitro* ubiquitination assays of WWP2, and we always see the autoubiquitination of WWP2 without substrate, but the authors did not detect any autoubiquitination signal. Thirdly, the change in ubiquitination signal is very subtotal based on the shown blot.

Response: Consistent with the observation by the reviewer, we also do detect WWP2 autoubiquitination in our experiments, although this is not constant (**Figure 1**). Of note auto-ubiquitination is either not visible in the *in vitro* assay by Yang, Y. *et al.* (*Proc Natl Acad Sci U S A* **110**, 5115-20, 2013, **Figure 2**), under similar conditions.

Although the p.R841H mutation is associated with increased TRIF ubiquitination *in vitro*, we agree that the change in subtotal and appears as a smear. This may be due to specific experimental conditions (time point, amount of reagent, type of gel etc.), as a reflect of the dynamic nature of the post-translational modifications.

Figure 1. *in vitro* ubiquitination assay of TRIF

[REDACTED]	[REDACTED] (Yang, Y. et al. Proc Natl Acad Sci U S A 110, 5115-20, 2013)
------------	---

Moreover, there is a huge gap between the activity change of WWP2 R841H and the biological consequences. If it is true that R841H mutation promotes TRIF ubiquitination, what are the biological effects? Does this mutation promote TRIF ubiquitination in the cell? What type of ubiquitination? Does ubiquitination leads to degradation or other effects? Why the enhanced ubiquitination leads to change in viral responses?

Response: in this paper, we identify a very rare variant in WWP2 in a 14 months old girl with HSE. Previous variant interfering with TLR3-IFN have been associated with the same disease, in children who otherwise do not develop severe infections. Furthermore, such variants have a low penetrance, suggesting that concomitant events at the time of infection (such as co-infection with another pathogen) are necessary for HSE to develop. This suggests that even subtle alterations in the biological process may lead to HSE.

It has been shown that WWP2 targets TRIF for K48-linked ubiquitination and degradation upon TLR3 activation, and that knockdown of WWP2 leads to increased TRIF protein level and an enhanced expression of IFN B (Yang, Y. *et al. Proc Natl Acad Sci U S A* **110**, 5115-20, 2013). Therefore, we hypothesized that the p841H WWP2 variant may affect the patient's ability to properly respond to HSV-1 infection.

Although our study does not provide the timeline and details of the ubiquitination process leading to the defect in TLR3 signaling, we believe that it reasonably supports this hypothesis. First, the mutation in WWP2 is very rare and not previously associated with a disease, second, it is clearly associated with impaired TLR3 signaling and enhanced HSV-1 susceptibility like observed in TRIF-deficient patients developing HSE, third, TLR3 signaling is rescued by the introduction of the WT allele in WWP2 p.R841H cells, and impaired by the introduction of mutant allele in WT cells. Nevertheless, in order to account for the studies' limitations, we have down toned the text when referring to the exact mechanism and remove the term "gain of function" from the title.

Reviewer #3 (Remarks to the Author):

The authors made some improvements as suggested by reviewers's
They should further build the case in regards to

- 1) molecular mechanism

Response: although our study does not provide the timeline and details of the ubiquitination process leading to the defect in TLR3 signaling, we believe that it reasonably supports this hypothesis, as discussed above (last point by reviewer 2).

- 2) reconstitution experiment (no statistics in Fig 9).

Response: according to the reviewer's comments, we provided statistics in panel G and H from previous Figure 9 (now Figure 8), as experiments are performed at 3 different times points (linear regression on log₁₀ + 1 transformed pfu). Experiments in panels A-F were performed in only 2 clones of CRIPR-Cas 9 modified cell lines, thereby preventing the ability to perform statistical analyses. However, we believe that it is important to show these data because the genetic modification was fully consistent in both directions (functional rescue after correction to WT and functional defect after introduction of the mutation in the WT).

3) demonstration of changes in ubiquitination in patients cells and/or cells expressing the mutant WWP2 protein (Figure 8 is insufficient)

Response: we agree that additional experiments, in particular ubiquitination assays in neuronal cells, would further support the study hypothesis. In order to account for the studies' limitations, we have down toned the text when referring to the exact mechanism and remove the term "gain of function" from the title.